# Developing a Digital Twin Framework for Wind Tunnel Testing: Validation of Turbulent Inflow and Airfoil Load Applications

Rishabh Mishra[1], Emmanuel Guilmineau[1], Ingrid Neunaber[2], and Caroline Braud[1]

[1]LHEEA lab. - CNRS - Nantes Université, Centrale Nantes, 1 rue de la Noë, 44100 Nantes, France
[2]NTNU, Høgskoleringen 1, 7034 Trondheim, Norway

**Correspondence:** Rishabh Mishra (rishabh.mishra@ec-nantes.fr)

**Abstract.** Wind energy systems, such as horizontal-axis wind turbines and vertical-axis wind turbines, operate within the turbulent atmospheric boundary layer, where turbulence significantly impacts their efficiency. Therefore, it is crucial to investigate the impact of turbulent inflow on the aerodynamic performance at the rotor blade scale. As field investigations are challenging, in this work, we present a framework where we combine wind tunnel measurements in turbulent flow with a digital twin of the experimental set-up. For this, first, the decay of the turbulent inflow needs to be described and simulated correctly. Here, we use Reynolds-Averaged Navier-Stokes (RANS) simulations with $k - \omega$ turbulence models, where a suitable turbulence length scale is required as an inlet boundary condition. While the integral length scale is often chosen without a theoretical basis, this study derives that the Taylor micro-scale is the correct choice for simulating turbulence generated by a regular grid: the temporal decay of turbulent kinetic energy (TKE) is shown to depend on the initial value of the Taylor micro-scale by solving the differential equations given by Speziale and Bernard (1992). Further, the spatial decay of TKE and its dependence on the Taylor micro-scale at the inlet boundary are derived. With this theoretical understanding, RANS simulations with $k - \omega$ turbulence models are conducted using the Taylor micro-scale and the TKE obtained from grid experiments as the inlet boundary condition. Second, the results are validated with excellent agreement with the TKE evolution downstream of a grid obtained through hot-wire measurements in the wind tunnel. Third, the study further introduces an airfoil in both the experimental and the numerical setting where 3D simulations are performed. A very good match between force coefficients obtained from experiments and the digital twin is found. In conclusion, this study demonstrates that the Taylor micro-scale is the appropriate turbulence length scale to be used as the boundary condition and initial condition to simulate the evolution of TKE for regular-grid-generated turbulent flows. Additionally, the digital twin of the wind tunnel can accurately replicate the force coefficients obtained in the physical wind tunnel.

## 1 Introduction

Wind energy systems, such as horizontal-axis and vertical-axis wind turbines, operate in a turbulent atmospheric boundary layer, which significantly affects their efficiency. Therefore, it is essential to study the turbulent inflow that they encounter. Important statistical quantities that considerably affect the aerodynamic performance of a rotor blade are the turbulent kinetic energy (TKE) and length scales in the wind. To study their effects, field experiments can be carried out, but they are complex,

time-consuming, and costly. Alternatively, experiments can be conducted in a wind tunnel by subjecting a Reynolds-scaled blade section from a real wind turbine blade to turbulent inflow under different inflow conditions, such as homogeneous inflow or gust inflow (see e.g., (Wei et al., 2019; Wester et al., 2022; Mishra et al., 2022; Nietiedt et al., 2022) and (Neunaber and Braud, 2020a)). However, wind turbines have grown to become the largest flexible, rotating machines in the world, with blade lengths approaching now 120 m. The interaction between a highly variable inflow and the unsteady aerodynamics of the moving and deforming blades is pushing the limits of current theory (Veers et al., 2019). At the blade scale, chord-based Reynolds numbers will exceed 15 million, which is unreachable in most available wind tunnels, and reach limits of pressurised facilities that are specifically developed for that purpose (Miller et al., 2019; Brunner et al., 2021). Therefore, having a digital twin model is key to help in the development of very large horizontal axis wind turbines (HAWT). Indeed, with these digital twin models, aerodynamic loads from local blade sections, used as input of Blade Element Momentum (BEM) solvers, can be pushed towards very large Reynolds numbers in numerous configurations encountered locally by blade sections (due to elasticity and floating movements). This would also provide the full flow and load description to understand further unsteady aerodynamics of blades in such configurations and thus further improve blade design of very large HAWT. Among the Computational Fluid Dynamics (CFD) methods available for that purpose, a Reynolds Average Navier-Stokes (RANS) formulation will be preferred over solving the Navier Stokes Equations directly with numerical methods using Direct Numerical Simulation (DNS) or Large Eddy Simulation (LES), as it is computationally too expensive for the targeted system. However, many challenges still remain and some of them will be tackled in this paper at low Reynolds number as a first step to be able to provide experimental validation. It concerns:

1. The theoretical description of the decay properties of turbulence measures such as the TKE and the dissipation rate in the downstream flow direction.

2. The numerical replication of the decay properties of turbulence measures such as the TKE and the dissipation rate in the downstream flow direction.

3. The reproduction of the aerodynamic loads acting on an airfoil by means of RANS simulations.

The literature review is therefore split into three sub-sections to give the reader an overview over decaying turbulence, focusing on the known turbulent flow properties behind grids, the great effort made in the literature to replicate these properties in simulated flows, and the impact of turbulence on the aerodynamics of an airfoil.

**Brief theoretical description of decaying grid turbulence**

The Navier-Stokes equations are nonlinear and their solutions are non-unique in nature. Therefore, simplifications are often made when describing or simulating turbulent flows, and here, we focus on literature on grid generated turbulence (GGT). Over the decades, GGT has been investigated intensely, and different works looked into the empirical and analytical description of the evolution of the turbulence decay. According to Batchelor and Townsend (1948), the decay of the turbulence intensity can be described by means of a power law. It has further been discussed that the inlet conditions play a role in the decay

of the turbulence (Kurian and Fransson, 2009). The works of Comte-Bellot and Corrsin (1966) show experimentally that the ratio between TKE ($k = \frac{1}{2}(\overline{u_1^2} + \overline{u_2^2} + \overline{u_3^2})$ with the velocity fluctuations $u_1$, $u_2$, and $u_3$ in three axis directions) and dissipation ($\epsilon = 30\nu\frac{\overline{u_1^2}}{\lambda^2}$, with the kinematic viscosity $\nu$ and the Taylor micro-scale $\lambda$) evolve linearly in the downstream direction. Krogstad

and Davidson (2009) showed that grid turbulence is Saffman turbulence (Saffman, 1967). They improved the decay exponent of TKE from 1.2, which Saffman gave for perfectly homogeneous, isotropic turbulence, to 1.1 for GGT. Sinhuber et al. (2015) performed experiments with one grid for different grid-mesh-size-based Reynolds numbers ($Re_M = \frac{UM}{\nu}$, where $M$ is the grid mesh size, and $U$ the mean velocity) and found that the decay exponent of the TKE was equal to 1.18. They also showed that the decay exponent was independent of $Re_M$. A literature review can be found in Kurian and Fransson (2009). We will use the

existing framework that was very briefly summarised above as a starting point for our theoretical framework, and a detailed description of both is given in section 2.

**Simulating decaying grid turbulence**

Simulating the inflow environment using computational fluid dynamics (CFD) has been the subject of many research works, and different turbulent formulations can be used that are summarized below from the most expensive computational effort to

the least one, i.e., RANS models.

Nagata et al. (2008) performed DNS to simulate the turbulent mixing layer for grid-generated turbulence, and they replicated typical GGT including the shear mixing layer. Continuing this work, Suzuki et al. (2010) showed that for a given mesh Reynolds number ($Re_M$), turbulent mixing is enhanced for fractal grids compared to regular grids. Laizet and Vassilicos (2011) simulated both a regular and different fractal grids using DNS, and they confirmed the characteristic regions of turbulence pro-

duction and decay downstream of fractal grids. Continuing their work, Laizet et al. (2013) studied inter-scale energy transfer of decaying turbulence for fractal grids. Efforts have also been made to use LES to simulate GGT. For example, Blackmore et al. (2013) developed a grid inlet technique to generate high-intensity turbulence for given length scales in LES simulations. With this, they successfully imitated the independence of the decay rate of the turbulence intensity (TI) from the mesh-size-based Reynolds number $Re_M$. Rieth et al. (2014) compared two LES models, namely the sigma model (Nicoud et al., 2011) and the

Smagorinsky model (Smagorinsky, 1963), to simulate grid-generated turbulence. They found that the sigma model is a good alternative to the static Smagorinky model and comparable to the dynamic Smagorinky model (Germano et al., 1991). Further, Liu et al. (2017) employed 3D LES to simulate GGT, and they found the same trend of absolute values of the mean TI. Djenidi (2006) used the Lattice Boltzmann method to simulate GGT, and they simulated the decay power law over a short distance from the grid but found it difficult to find the decay exponent.

Finally, Torrano et al. (2015) have investigated the performance of various two-equation eddy viscosity models for predicting the decay of the turbulent kinetic energy downstream of a regular grid by means of RANS. They calculated the dissipation rate using an integral length scale equal to the grid mesh size. The simulation results were compared with experimental results, and a conclusion of their work was that eddy viscosity models were over-predicting the turbulent kinetic energy in comparison to the experiments. As an alternative to two-equation eddy viscosity models, Reynolds stress transport models (RSTM) can

also be used where transport equations for each of the Reynolds stress terms are solved. For example, Panda et al. (2018)

performed RANS simulations using RSTM. However, because of the closure problem, using RSTM is quite complex as the nine non-closed components of the nine Reynolds stress transport equations have to be modeled.

Regarding RANS equations, studies focus mostly on improving the eddy viscous model, while not much effort was put on the choice of the length scale that is usually assumed to have an order of magnitude of the integral length scale. Two quantities are generally used at the inlet of RANS simulations: the turbulence intensity and the length scale. Indeed, when using the most popular two-equation eddy viscosity models, namely the $k - \epsilon$ and $k - \omega$ models (Wilcox, 1988), or a mixture of these two models, the $k - \omega$ SST Menter 1994 and 2003 models (Menter, 1994; Menter et al., 2003), the following quantities need to be provided at the inlet boundary: $k$, the turbulent kinetic energy, $\epsilon$, the dissipation rate, and $\omega$, the specific dissipation rate. $k$ is generally given through a turbulence intensity quantity, and $\epsilon$ and $\omega$ are computed using $k$ and a given characteristic turbulence length scale chosen individually for each flow configuration. For example, in the case of pipe flow, the turbulence length scale is empirically approximated as 0.07 times the diameter of the pipe (Versteeg and Malalasekera, 1995).

We would like to emphasize that a correctly evaluated and properly defined turbulence length scale is a basic requirement to accurately capture the evolution of turbulence. Finally, in principle, if provided with an adequate closure, the RANS model should be able to capture statistical measures of turbulence and their evolution in the flow field. The present work will focus on improving the length scale choice to reproduce the correct evolution of $k$.

**Experimental and numerical aerodynamic research**

As stated above, the third challenge of the work presented here is the reproduction of aerodynamic loads on a 2D airfoil section by means of RANS simulations in particular for low Reynolds numbers. There are many experimental works that describe the evolution of the lift and drag coefficients with the variation of the angle of attack (AoA) of the airfoil, and a summary of the design and aerodynamics for wind turbine blades can for example be found in Bak (2022). For this work, the impact of low Reynolds numbers and the impact of turbulence on the performance of 2D blade sections is of interest. Different studies on the influence of the turbulence intensity in the inflow on the aerodynamics of a 2D blade section are summarized in Li and Hearst (2021). There, an NREL S826 profile is exposed to different turbulence levels up to 5.4% for a chord-based Reynolds number of $Re_c = 4 \cdot 10^5$ defined as $Re_c = \frac{c \cdot U_\infty}{\nu}$, where $c$ denotes the chord length, $U_\infty$ denotes the inflow velocity, and $\nu$ the kinematic viscosity. They found that the slope of linear part of the lift curve increases with increasing turbulence intensity and that turbulence intensities of up to $1.6\%$ led to a reduction of the maximum lift whereas turbulence intensities of $2.1\%$ and higher led to an increase in lift compared to the reference case. Devinant et al. (2002) and Sicot et al. (2006) studied the influence of turbulence intensities up to 16% on the aerodynamics of a 2D NACA65(4)421 blade section. The chord-based Reynolds numbers were $1 \cdot 10^5 \leq Re_c \leq 7 \cdot 10^5$. Devinant et al. (2002) showed that increasing the turbulence intensity shifts the stall angle towards higher angles of attack. This is attributed to a turbulent boundary layer flow that is known to be less prompt to flow separation, which also displaces the transition towards the trailing edge. Sicot et al. (2006) found that the fluctuations of the surface pressure measurements, characterized by the standard deviation, increased in the separated flow region with increasing turbulence intensity. The average location of the separation line was not affected.

Simulating turbulent flow upstream of the airfoil often requires the use of LES, as demonstrated in the work of Gilling et al. (2009). In this paper, we will show that with appropriate boundary conditions at the inlet, RANS simulations can yield an accurate evolution of turbulence properties.

Finally, fewer investigations of the impact of turbulence have been performed at higher Reynolds numbers due to experimental complexity. The present digital-twin model will pave the way for such studies.

**Structure of this paper**

As stated above, the aim of this work is the creation of a digital twin of a low-Reynolds number wind tunnel where turbulence is generated with grids. For this, first, a theoretical framework is developed in section 2 to demonstrate that the Taylor micro-scale ($\lambda$) is the correct length scale to be used as the turbulence length scale in RANS simulations for homogeneous, isotropic turbulent flows. Additionally, we show that the spatial and temporal decays of turbulent kinetic energy are directly dependent

on the Taylor micro-scale, and we derive a relation between the Taylor micro-scale and the downstream position. In section 3, the theoretical framework is then validated using RANS simulations and experiments behind a homogeneous grid; the experimental and numerical set-ups are detailed in sections 3.1 and 3.2. A validation and the results are presented in section 3.4. In the next step, the digital twin is used to perform aerodynamic simulations which are compared to experiments. The numerical and experimental set-ups are explained in sections 4.2 and 4.1, and the results are compared in section 4.3. Finally,

conclusions and perspectives are given in section 5.

## 2   Theoretical framework

In the following, we will detail the theoretical framework of the decay of homogeneous, isotropic turbulence that will lay the foundation of our digital twin. We will start with some important equations from literature and develop them further.

### 2.1   Dependence of the temporal evolution of $k$ on $\lambda$

Researchers in the physics and mathematics community have studied the case of homogeneous isotropic turbulence in great detail. Batchelor and Townsend (1948) proposed to call the Reynolds number based on the Taylor micro-scale ($\lambda$) the "Reynolds number of turbulence", and also suggested that $\lambda$ is representative of the eddies of large wave-number, i.e., small eddies, before viscosity becomes relevant. The mathematical definition and the experimental methodology for calculating $\lambda$ are detailed in section 3.3. In the theory of homogeneous isotropic turbulence, it is necessary to assume the similarity of turbulence at all

stages of decay, i.e., changes in the structure of turbulence can be described by two parameters, namely a characteristic length and a characteristic velocity (Stewart and Townsend, 1951). Later on, George (1992) proved that the characteristic length scale of the entire energy spectrum is the Taylor micro-scale. This was confirmed by Speziale and Bernard (1992), who also proposed that the turbulent kinetic energy and the Taylor micro-scale are the appropriate scaling parameters for all scales of motion. They proposed a set of differential equations to calculate the temporal decay of turbulent kinetic energy, $k(t)$, and dissipation rate,

$\epsilon(t)$ :

$$\frac{dk(t)}{dt} = \epsilon(t), \tag{1}$$

$$\frac{d\epsilon(t)}{dt} = -\alpha \frac{\epsilon(t)^2}{k(t)}. \tag{2}$$

Here, $\alpha$ is a constant. Equations (1) and (2) formulate an initial value problem which can easily be solved to give a temporal decay law for initial conditions at time $t = 0$, $k(t) = k(0)$ and $\epsilon(t) = \epsilon(0)$, cf. (Zhou and Speziale, 1998), which gives

$$k(t) = k(0) \left( 1 + \frac{1}{\alpha} \frac{\epsilon(0)}{k(0)} t \right)^{-\alpha}. \tag{3}$$

With the equation from Bailly and Comte-Bellot (2015),

$$\frac{k(0)}{\epsilon(0)} = \frac{(\lambda(0))^2}{20\nu}, \tag{4}$$

we can write equation (3) as

$$k(t) = k(0) \left( 1 + \frac{1}{\alpha} \frac{20\nu}{(\lambda(0))^2} t \right)^{-\alpha}. \tag{5}$$

Looking at equation (5), we can clearly say that the temporal decay of the turbulent kinetic energy has a direct dependence on the initial Taylor micro-scale $\lambda(0)$. This is comforted by Mydlarski and Warhaft (1996) who empirically found a relationship between the energy decay exponent and the Taylor Reynolds number $R_\lambda$.

## 2.2 Dependence of the spatial evolution of $k$ on $\lambda$

The dependence of the downstream evolution of $k$ on $\lambda$ is demonstrated here within the framework of homogeneous, isotropic turbulence. For the steady-state case, the transport equation for $k$ can be written as (here, we follow Einstein's summation convention):

$$U_j \frac{\partial k}{\partial x_j} = \frac{\partial}{\partial x_j} \left( \frac{1}{\rho} \overline{u_j p} + \frac{1}{2} \overline{u_i u_i u_j} - 2\nu \overline{u_i s_{ij}} \right) - \overline{u_i u_j} S_{ij} - \epsilon. \tag{6}$$

$U_j$ and $u_j$ are the mean velocity and the velocity fluctuation, respectively, with $i, j = 1, 2, 3$. The pressure fluctuations are

denoted by $p$, and $\nu$ is the kinematic viscosity of the fluid. $s_{ij}$ is the strain rate tensor for velocity fluctuations, and $S_{ij}$ is the strain rate tensor for the mean flow, defined as

$$S_{ij} = \frac{1}{2} \left( \frac{\partial U_i}{\partial x_j} + \frac{\partial U_j}{\partial x_i} \right), \tag{7}$$

$$s_{ij} = \frac{1}{2} \left( \frac{\partial u_i}{\partial x_j} + \frac{\partial u_j}{\partial x_i} \right). \tag{8}$$

The turbulent kinetic energy is defined as

$$k = \frac{1}{2}(\overline{u_1^2} + \overline{u_2^2} + \overline{u_3^2}). \tag{9}$$

It should be noted that for grid-generated turbulence

$$\overline{u_1^2} = 1.2\overline{u_2^2} = 1.2\overline{u_3^2} = \overline{u^2} \tag{10}$$

cf., (Comte-Bellot and Corrsin, 1966; Bailly and Comte-Bellot, 2015).

After substituting (10) in (9), we get

$$k = \frac{4}{3}\overline{u^2}. \tag{11}$$

If the dominant flow direction is the $x_1$ direction, then equation (6) can be written as

$$U_1 \frac{\partial k}{\partial x_1} = \underbrace{\frac{\partial}{\partial x_1}\left(\frac{1}{\rho}\overline{u_1 p} + \frac{1}{2}\overline{u_i u_i u_1} - 2\nu\overline{u_i s_{i1}}\right)}_{transport} - \underbrace{\overline{u_i u_1}S_{i1}}_{production} - \epsilon. \tag{12}$$

If we assume the turbulence to be homogeneous and decaying, then the term representing the transport of $k$ in an inhomogeneous field due to pressure fluctuation, the turbulence itself, and viscous stresses, and the term representing the production of $k$ can be set to zero (Bailly and Comte-Bellot, 2015).

Therefore, equation (12) becomes

$$\frac{\partial k}{\partial x} = -\epsilon/U. \tag{13}$$

Here, we have dropped the indices for the sake of simplification. For homogeneous, isotropic turbulence, the dissipation rate $\epsilon$ can be related to both the longitudinal Taylor micro-scale ($\lambda_1$) and the transverse Taylor micro-scale ($\lambda_2$) as

$$\epsilon = 15\nu\frac{\overline{u^2}}{\lambda_2^2} = 30\nu\frac{\overline{u^2}}{\lambda_1^2}. \tag{14}$$

By using equation (11), we can write equation (14) as

$$\epsilon = 11.25\nu\frac{k}{\lambda_2^2} = 22.5\nu\frac{k}{\lambda_1^2}. \tag{15}$$

From here onwards, we will only use the relation corresponding to $\lambda_1$, and for simplicity, we will drop the subscript '1'. Hence, the relation for the dissipation rate $\epsilon$ can be written as

$$\epsilon = 22.5\nu \frac{k}{\lambda^2}. \tag{16}$$

By substituting $\epsilon$ from equation (16) into equation (13), we arrive at

$$\frac{\partial k}{\partial x} = -\frac{22.5\nu}{U} \frac{k}{\lambda^2}. \tag{17}$$

Since the evolution of $k$ is only a function of $x$, for the given case, the partial differential equation (17) becomes an ordinary differential equation,

$$\frac{dk}{dx} = -\frac{22.5\nu}{U} \frac{k}{\lambda^2}. \tag{18}$$

Solving the differential equation (18),

$$k(x) = C_1 exp\left(-\frac{22.5\nu}{U} \int \frac{1}{\lambda^2}\, dx\right). \tag{19}$$

Equation (19) shows that the evolution of $k$ in the downstream direction has a direct dependence on the Taylor micro-scale $\lambda$. Here, $C_1$ is a constant.

To find the final solution, we need to understand the dependence of $\lambda$ on $x$. This is performed in the following section.

## 2.3 Dependence of $\lambda$ on $x$

From equation (16) we know that $\lambda^2 \propto k/\epsilon$. Experiments have shown that for homogeneous isotropic turbulence, $k/\epsilon$ evolves linearly in the downstream direction, e.g., (Comte-Bellot and Corrsin, 1966). Thus, we can write

$$\frac{k}{\epsilon} = \frac{k_{in}}{\epsilon_{in}} + \frac{m}{U}x, \tag{20}$$

where $m/U$ is the slope. Now, using equations (16) and (20), we can write

$$\lambda^2 = 22.5\nu\left(\frac{k}{\epsilon}\right) = 22.5\nu\left(\frac{k_{in}}{\epsilon_{in}} + \frac{m}{U}x\right). \tag{21}$$

$k_{in}$ and $\epsilon_{in}$ are the values of the turbulent kinetic energy and the dissipation rate at the starting point $x = 0$ from where the TKE's decay is calculated. Substituting equation (21) into equation (19), performing integration, and subjecting it to the boundary condition at the starting point, $k(x = 0) = k_{in}$, gives

$$k(x) = k_{in}\left(1 + m\frac{\epsilon_{in}}{k_{in}}\frac{x}{U}\right)^{-1/m}. \tag{22}$$

Using relation (16), we can rewrite equation (22) as

$$k(x) = k_{in}\left(1 + m\frac{22.5\nu}{\lambda_{in}^2}\frac{x}{U}\right)^{-1/m}, \tag{23}$$

were $\lambda(x = 0) = \lambda_{in}$. It should be noted that for equation (23), the starting point, i.e., $x = 0$, and, thus, the origin of the coordinate system, lies at the point of measurement which is used to give the value of $k_{in}$ and $\lambda_{in}$. Looking at equation (23), we can say that the evolution of $k$ in the downstream direction can be defined for a homogeneous, isotropic flow, provided we have the values of $k_{in}$ and $\lambda_{in}$ at one point.

From the framework of the $k - \omega$ series of models for RANS simulations, one can also derive the following evolution equation for $k$ for decaying, homogeneous, isotropic turbulence (Eça et al., 2016),

$$k(x) = k_{in}\left(1 + \omega_{in}\beta\frac{x}{U}\right)^{-\beta^*/\beta}, \tag{24}$$

where $\omega$ is the so-called turbulence frequency that is related to $\epsilon$ and $k$ by

$$\omega = \frac{\epsilon}{\beta^* k}, \tag{25}$$

and $\beta$ and $\beta^*$ are constants with values 0.0828 and 0.09 respectively (Wilcox, 1988). The value of $\omega$ at the inlet boundary is given by $\omega_{in}$. By substituting equation (25) into equation (24), we have

$$k(x) = k_{in}\left(1 + \frac{\beta}{\beta^*}\frac{\epsilon_{in}}{k_{in}}\frac{x}{U}\right)^{-\beta^*/\beta}. \tag{26}$$

Using relation (16), we can rewrite equation (27) as following,

$$k(x) = k_{in}\left(1 + \frac{\beta}{\beta^*}\frac{22.5\nu}{\lambda_{in}^2}\frac{x}{U}\right)^{-\beta^*/\beta}. \tag{27}$$

Equations (23) and (27) have a similar form. Even if the equation (27) comes from the $k - \omega$ model, believing in its proven applicability, we can assume that,

$$m = \frac{\beta}{\beta^*}. \tag{28}$$

Therefore, the value of $m$ can be taken as 0.92. It is important to notice that further investigations are required to find the correct value of the parameter $m$ theoretically. The authors wish to emphasise to the readers that, unlike the equations commonly

encountered in prior literature, such as those referenced in Comte-Bellot and Corrsin (1966), Kurian and Fransson (2009), Krogstad and Davidson (2009), or Sinhuber et al. (2015), equation (23) does not have any fitting parameter, and it is neither an empirical equation nor does it have any virtual origin. The only assumption taken while deriving the equation (23) is statistical stationarity of fully developed GGT which is believed to start from $x/M \approx 20$ downstream of a grid (Comte-Bellot and Corrsin, 1966; Bailly and Comte-Bellot, 2015). Upstream of $x/M \approx 20$, one may expect some changes in the form of equations (10), (13), and (20) which are used to derive equation (23). However, between $x/M \approx 10$ and $x/M \approx 20$, the turbulent flow can be considered approximately developed (Frisch, 1995). Therefore, for all practical purposes, equation (23) is valid downstream of $x/M \approx 10$. For the cases where the assumption of Taylor's hypothesis is valid, i.e., time $t$ equals $x/U$, the spatial evolution equation of $k$, equation (23), becomes equivalent to the temporal decay equation of $k$, cf. equation (5). Hence, measurements of $k$ and $\lambda$ at a given position in a regular-grid-generated turbulent flow will enable us to obtain the evolution of $k$ when solving the RANS equations. This is essential to reproduce realistic regular-grid-generated turbulent inflow conditions at the inlet boundary of RANS simulations. The experimental and numerical set-ups used to validate the spatial evolution equation (23) are presented in the following section.

## 3   Digital twin: simulating regular grid inflow in the wind tunnel

In the following, $k$ and $\lambda$ will be measured from the flow behind a grid in a wind tunnel facility. The objective is to demonstrate that the evolution of $k$ obtained from RANS simulations matches the experiments when using $k$ and $\lambda$ from measurements at a given location. The wind tunnel facility and the grid set-up are described below together with the RANS simulations performed.

### 3.1   Experimental set-up

The experiments were performed in the aerodynamic closed-loop low-Reynolds-number wind tunnel facility at the LHEEA (Laboratoire de recherche en Hydrodynamique, Énergetique et Environnment Atmospheric) laboratory of CNRS and Centrale Nantes (see figure 1). This wind tunnel has a cross section of 0.5 m × 0.5 m and a test section length of 2.3 m with a maximum inflow velocity of 40 ms$^{-1}$ and a turbulence intensity of less than 0.3%.

To induce a turbulent inflow, a regular wooden grid with square bars was used. The cross-section of the bars used for the frame is 11 mm × 10.5 mm and of the ones used for creating the mesh $d = 6$ mm × 6 mm. The blockage ($b$) of the grid is 16% and the grid mesh size $M$ is 70 mm × 70 mm.

To measure the downstream evolution of turbulence properties of the inflow, a 1D hot-wire probe of type 55P11 with a wire length of 1.25 mm from Dantec Dynamics was used. It was operated using a DISA55M01 unit. The hot-wire was calibrated in the velocity range $0.5 \, \mathrm{ms}^{-1} \leq U \leq 40 \, \mathrm{ms}^{-1}$ applying the temperature correction suggested by Hultmark and Smits (2010). During the measurements, the mean velocity was approximately 25 ms$^{-1}$. The data was recorded at a sampling frequency of 25 kHz for 10 seconds for calibration and 20 seconds for the measurements. A hardware low-pass filter with a cut-off frequency of 10 kHz was used. The downstream evolution of $k$ was measured at the centre-line of the wind tunnel at nine downstream positions that are indicated in figure 1.

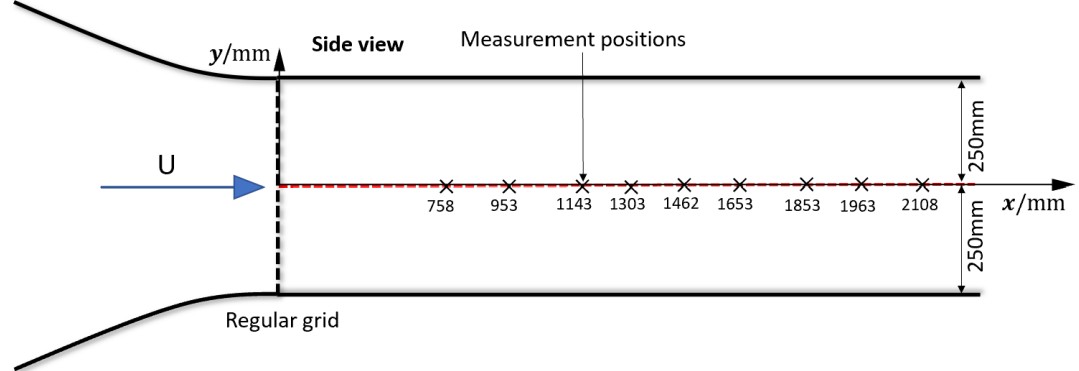

**Figure 1.** Schematic diagram of the experimental set-up for hot-wire measurements in the wake of a regular grid (side view). The measurement positions are marked.

## 3.2 Numerical set-up

The simulations are performed using the finite-volume-based ISIS-CFD incompressible URANS solver. This solver, developed by CNRS and Centrale Nantes, also available as a part of the FINE™/Marine computing suite worldwide distributed by Cadence Design Systems, uses an incompressible unsteady Reynolds-averaged Navier-Stokes (URANS) method. The solver is based on a finite volume method to build the spatial discretization of the transport equations. The unstructured discretization is face-based, which means that cells with an arbitrary number of arbitrarily shaped faces are accepted. A second order backward difference scheme is used to discretise time. All flow variables are stored at the geometric center of arbitrarily shaped cells. Volume and surface integrals are evaluated with second-order accurate approximations. As the method is face-based, numerical fluxes are reconstructed on the mesh faces by linear extrapolation of the integrand from the neighbouring cell centers. A centered scheme is used for the diffusion terms, whereas for the convective fluxes, a blended scheme with 80% central and 20% upwind is used. In the case of turbulent flows, additional transport equations for the variables in the turbulence model are added. In the following, the transport equation for URANS simulations are presented:

Momentum conservation equation:

$$\rho\frac{\partial\overrightarrow{U}}{\partial t} + \rho(\overrightarrow{U}\cdot\overrightarrow{\nabla}\overrightarrow{U}) = \rho\overrightarrow{g} - \overrightarrow{\nabla}\overline{p} + \mu\triangle\overrightarrow{U} - \rho\overrightarrow{\nabla}\cdot\overline{\overline{R}}. \tag{29}$$

Continuity equation for the mean component:

$$\overrightarrow{\nabla}\cdot\overrightarrow{U} = 0. \tag{30}$$

Continuity equation for fluctuations:

$$\overrightarrow{\nabla}\cdot\overrightarrow{u} = 0. \tag{31}$$

The Reynolds stress tensor $\overline{\overline{R}}$ present in the equation (29) is defined as

$$\overline{\overline{R}} = \mu_t \left( \overrightarrow{u} \otimes \overrightarrow{u} \right), \tag{32}$$

where $\overrightarrow{U}$ and $\overrightarrow{u}$ are the mean and fluctuating component of the velocity, and $\mu_t$ is the turbulent viscosity. There are many ways to calculate the value of $\mu_t$ and in this paper, a two-equation eddy viscosity method, namely $k - \omega$ SST Menter (2003), has been used to do so. In the following, a description of this model is given.

Transport equation for the turbulent kinetic energy $k$:

$$\frac{\partial(\rho k)}{\partial t} + \frac{\partial(\rho U_i k)}{\partial x_i} = \tilde{P}_k - \beta^* \rho k \omega + \frac{\partial}{\partial x_i}\left[(\mu + \sigma_k \mu_t)\frac{\partial k}{\partial x_i}\right]. \tag{33}$$

Transport equation for the turbulence frequency $\omega$:

$$\frac{\partial(\rho\omega)}{\partial t} + \frac{\partial(\rho\bar{u}_i\omega)}{\partial x_i} = \alpha\rho S^2 - \beta\rho\omega^2 + \frac{\partial}{\partial x_i}\left[(\mu + \sigma_\omega\mu_t)\frac{\partial\omega}{\partial x_i}\right] + 2(1 - F_1)\rho\sigma_{w2}\frac{1}{\omega}\frac{\partial k}{\partial x_i}\frac{\partial\omega}{\partial x_i}. \tag{34}$$

$F_1$ is the blending function. $F_1$ goes to zero for the flow away from the surface, hence the $k - \epsilon$ model is applied there and it goes to one near the wall where the $k - \omega$ model is applied. The constant $\alpha$ is computed by $\alpha = \alpha_1 F_1 + \alpha_2(1 - F_1)$. For this model, the values of constant are $\beta^* = 0.09$, $\alpha_1 = 5/9$, $\beta_1 = 3/40$, $\sigma_{k1} = 0.85$, $\sigma_{\omega1} = 0.5$, $\alpha_2 = 0.44$, $\beta = 0.0828$, $\sigma_{k2} = 1$, $\sigma_{\omega2} = 0.856$. Using these constants, both transport equations are solved and the turbulent viscosity is found by equation (35).

$$\mu_t = \frac{\rho a_1 k}{max(a_1\omega, SF_2)}, \tag{35}$$

where $S$ is the measure of the strain rate and $F_2$ is the second blending function defined by:

$$F_2 = \tanh\left[\left[max\left(\frac{2\sqrt{k}}{\beta^*\omega y}, \frac{500\nu}{y^2\omega}\right)\right]^2\right]. \tag{36}$$

Using the above-presented set of equations, 2D simulations are performed for the domain, mimicking the wind tunnel shown in figure 2, with a domain length of 3.12 m, a width of 0.5 m (same as the width of the wind tunnel test section), and 8000 cells. Simulations were performed with a higher number of cells (10000 and 12000) as well, but no changes were observed. The top and the bottom walls in the simulations were put to the no-slip condition.

It should be noted that the values put as the boundary condition at the inlet boundary are those obtained at the first point of the measurement, 758 mm downstream of the wind tunnel inlet which refers to $x/M = 11$ in figures 4 and 6.

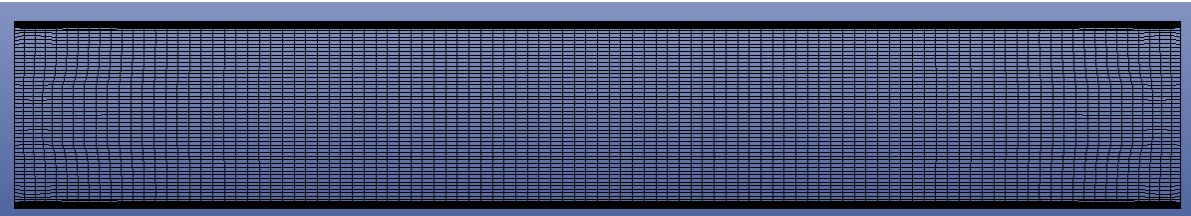

**Figure 2.** Simulation domain for decaying turbulence.

## 3.3 Length scales

For homogeneous, isotropic turbulence, both the integral length ($L$) and $\lambda$ can be easily estimated using the one-dimensional energy spectrum (Hinze, 1975). $L$ can be obtained by taking the limit of the energy spectrum $E(f)$ in the frequency domain for $f \to 0$,

$$L = \lim_{f \to 0} \left( \frac{E(f) \cdot U}{4\sigma^2} \right). \tag{37}$$

Here, $U$ is the mean streamwise velocity, $u$ denotes the fluctuations, and $\sigma$ is the standard deviation of the velocity time-series $U(t)$. It should be noticed that here that only the frequency range where $E(f) \approx const$ just outside of the inertial sub-range is used to determine $L$.

$\lambda$ is defined as following,

$$\lambda = \left( \frac{\sigma^2}{\langle (\frac{\partial u}{\partial x})^2 \rangle} \right)^{1/2}, \tag{38}$$

where $\langle (\frac{\partial u}{\partial x})^2 \rangle$ can be determined from the spectrum in the wave-number ($\kappa$) domain, derived through hot-wire measurements using Taylor's hypothesis,

$$\left\langle \left( \frac{\partial u}{\partial x} \right)^2 \right\rangle = \int\limits_{\kappa_{min}}^{\kappa_{max}} \kappa^2 E(\kappa) \, d\kappa, \tag{39}$$

where $\kappa_{min}$ and $\kappa_{max}$ are the wave number boundaries of the whole energy spectrum. At the downstream position $x/M = 11$, which represents the first point of measurement in the wind tunnel, we calculated $L/M$, $\lambda/M$, and the Taylor Reynolds number ($Re_\lambda$) using hot-wire measurements and equations (37) and (38). An exemplary spectrum is shown in figure 3. The calculated values were $L/M = 0.39$, $\lambda/M = 0.034$, and $Re_\lambda = 180$, respectively.

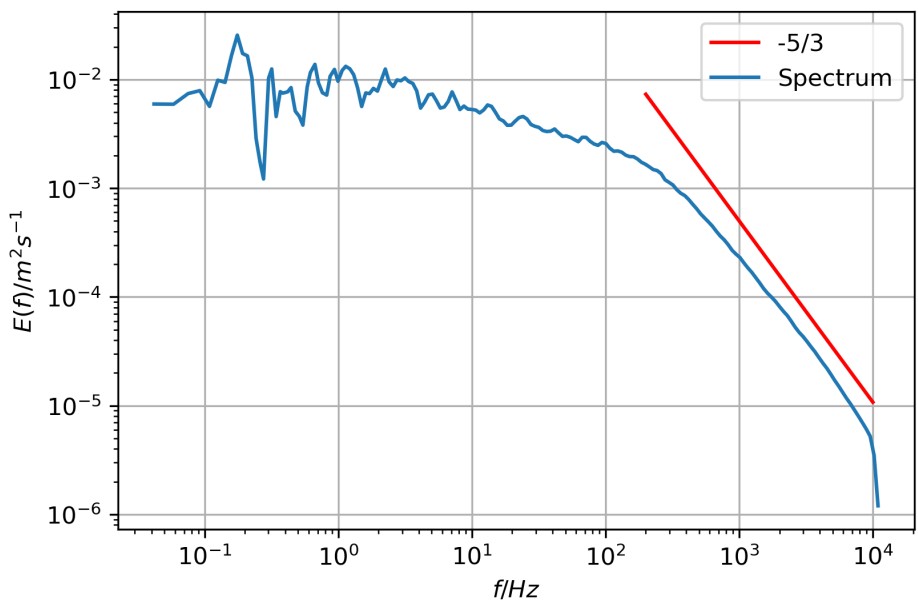

**Figure 3.** Power spectrum obtained from hot-wire measurements of the flow downstream of the regular grid in the frequency domain for the position $x/M = 11$.

### 3.4 Validation

Figure 4 shows the downstream evolution of $k$ obtained theoretically with equation (23), experimentally using hot-wire measurements (section 3.1), and from simulation using the $k - \omega$ SST Menter 2003 model (see section 3.2) with both $L$ and $\lambda$ as boundary conditions at the inlet. First, we see that the theoretical equation is validated against experiments. The log-log equivalent of the curve is provided in the inset. Next, simulation results obtained using both the Taylor micro-scale and the integral length as the boundary condition at the inlet are compared with equation (23). Here, we can clearly see that the simulation result obtained for the case where the Taylor micro-scale is used as the boundary condition matches very well with the theory. The TKE's decay exponent was determined to be 1.087 in the equation (23) and simulation, while in the experimental data, it was observed to be 1.09. These values are in close proximity to the Saffman decay exponent of 1.1 for grid-generated turbulence, as reported by Krogstad and Davidson (2009). In contrast, when the integral length is used as boundary condition, the simulation results do not match the theory or the experimental results. The justification for this observation comes directly from equation (23) where the derived 1D spatial evolution equation shows a direct dependence of the turbulent kinetic energy on the Taylor micro-scale.

To verify the generality of equation (23), apart from the validation performed against the TKE decay data obtained in the LHEEA wind tunnel, we also compared it with the results from hot-wire experiments conducted independently in the wind tunnel at the University of Oldenburg and data given in Batchelor and Townsend (1948). The Oldenburg experiments were per-

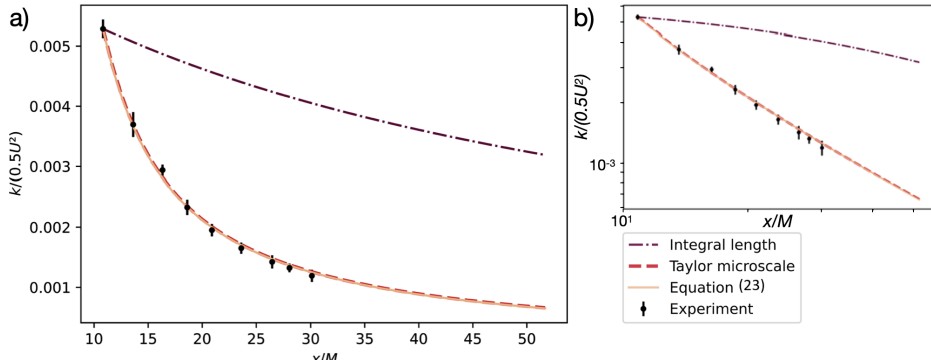

**Figure 4.** Comparison of the downstream evolution of $k/(0.5U^2)$ obtained from theory with experiments, and simulations performed with integral length and Taylor micro-scale as the inlet boundary condition using the $k-\omega$ SST Menter 2003 model. (a) linear; (b) logarithmic.

formed using a passive regular grid with $M = 115$ mm for 33 downstream positions spanning from $x/M \sim 8$ to $x/M \sim 170$ for two inflow speeds: 5 ms$^{-1}$ and 10 ms$^{-1}$. Figures 5 shows the log-log plot of the comparison of the TKE decay obtained experimentally with that from equation (23). It can clearly be seen that the evolution of TKE given by equation (23) matches very well with the experimental data. Note that the deviations are over-accentuated when visualised in the log-log plot. Readers interested in knowing the details of the experiments performed at the University of Oldenburg may refer to appendix A.

It is also interesting to see how different $k-\omega$ models calculate the downstream evolution of $k$ by using $\lambda$ as the boundary condition. This is shown in figure 6. Here, we can see that the results from the $k-\omega$ SST Menter 2003, $k-\omega$ SST Menter 1994 and $k-\omega$ BSL Menter overlap with only a little deviation for the oldest $k-\omega$ series of the turbulence models, the $k-\omega$ Wilcox model.

This emphasises the very important role of the choice of the turbulent length scale at the simulation domain inlet as compared to the choice of the turbulent model. The following section showcases an instance of the digital twin's functionality, where an airfoil is subjected to testing in both the physical wind tunnel and the digital twin, with a subsequent comparison of the resulting loads.

## 4 Example of the performance of the digital twin: testing an airfoil section of a wind turbine

The methodology to obtain a digital twin model of the experimented turbulent inflows is now validated. The present section is focusing on modeling the wind tunnel experiments to reproduce loads when a turbulent inflow is set. The same turbulent inflow as described in section 3.1 is used, which induces a homogeneous field with a turbulence intensity of 3%, measured at the airfoil position before its installation.

We are then introducing an airfoil model described in section 4.1.1, which is equipped with global load sensors (section 4.1.2) and local pressure sensors (section 4.1.3).

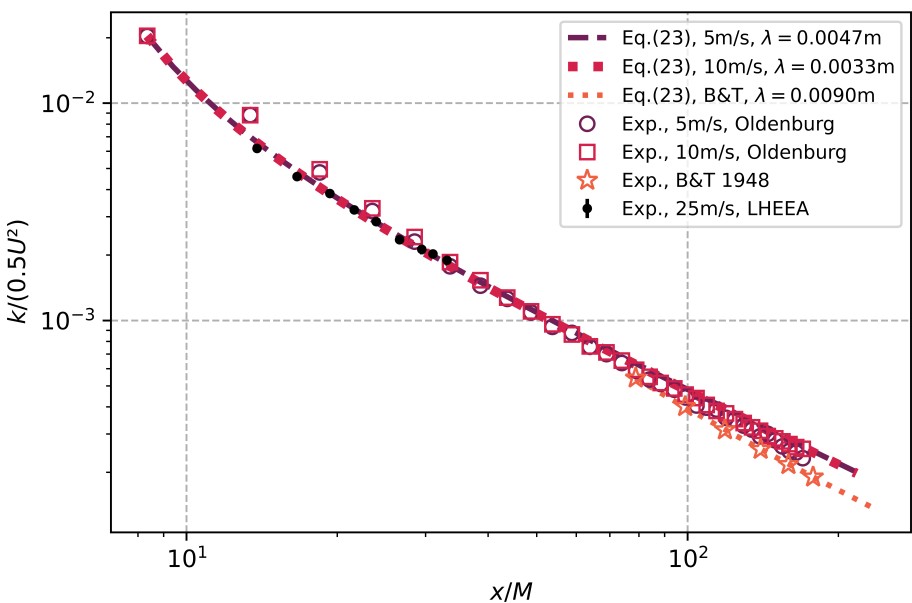

**Figure 5.** Comparison of experimental decay of normalised TKE decay obtained from experiments performed in the University of Oldenburg, and data from Batchelor and Townsend (1948) (B&T) with equation (23). For the application of equation (23) to the B&T data, we estimate $\lambda \approx 9$ mm from their manuscript.

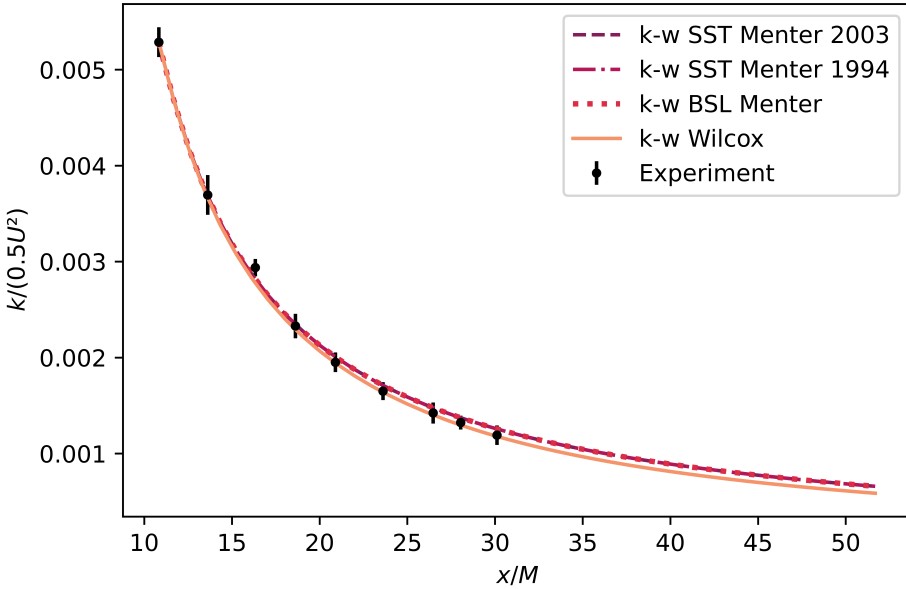

**Figure 6.** Comparison of the downstream evolution of $k/(0.5U^2)$ computed using different $k - \omega$ models

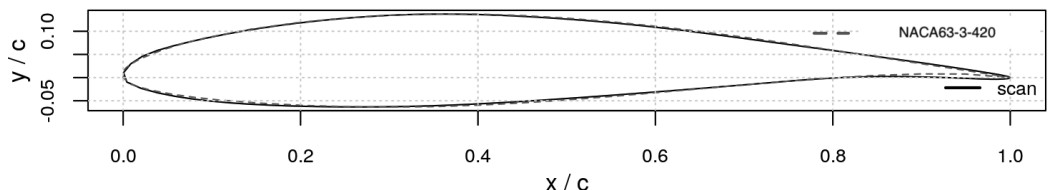

**Figure 7.** Blade section at 82% of the radius in comparison with a NACA63-3-420 profile with a modified camber of 4% instead of 2% (from Neunaber et al. (2022); published under CC BY-NC-ND 4.0 (License)).

## 4.1 Experimental set-up

### 4.1.1 Airfoil

A 2D blade section was placed 1.79 m downstream of the wind tunnel inlet where the flow field is homogeneous (Neunaber and Braud, 2020b) in the wind tunnel described in section 3.1. The inflow velocity is 25 ms⁻1. The airfoil shape was derived from scans of a 2MW wind turbine blade section, at 82% of its length (Neunaber et al., 2022). It has been scaled-down to 1/10th of the original chord length, so that the chord length is $c = 0.125$ m and the chord-based Reynolds number is $Re_c = 2.0 \times 10^5$.

The airfoil section closely resembles a NACA63-3-420 profile with a modified camber of 4% instead of 2% (see figure 7). The 2D blade section has been designed with multiple sensors to perform a 3D characterisation of the wall pressure over the airfoil surface, and future actuators and/or sensors can be implemented on the suction side (see figure 8). It was manufactured from aluminum using a 3D metal printer to integrate channels for the pressure measurements. In total, four pressure scanners with sixteen channels each are used, and the locations of the pressure taps are given in section 4.1.2. The model is hollow

and equipped with four covers on the suction side for access to the sensors. This also allows for the integration of actuators in the future. To perform simulations with the digital twin, it was important to check the shape of the airfoil for deviations and unevenness after manufacturing. Therefore, the down-scaled model has been scanned using the HandyScan 3D 700$^{MC}$ from CREAFORM that has an accuracy of 0.03 mm. The scan was then compared to the initially designed shape that is used in simulations (see figure 9). At first, the covers of the 2D blade section had steps as high as 0.7 mm, which was not accurate

enough to produce experimental results that matched the simulations. These steps were significantly improved manually to an accuracy of less than 0.45 mm, which was sufficient to match simulation results. It should be noted that for this inflow velocity, a TI of at least 3% was necessary to avoid low Reynolds number effects found in previous investigations (see Mishra et al. (2022)).

### 4.1.2 Local pressure sensors

The blade section is equipped with four differential pressure scanners from EvoScann® (P-Series) with 16 channels each that have a range of ±50 mbar. The acquisition rate can be up to 1 kHz, but it was limited to 100 Hz for the present measurements as only steady quantities were targeted. These sensors were connected to wall pressure taps through channels integrated in the

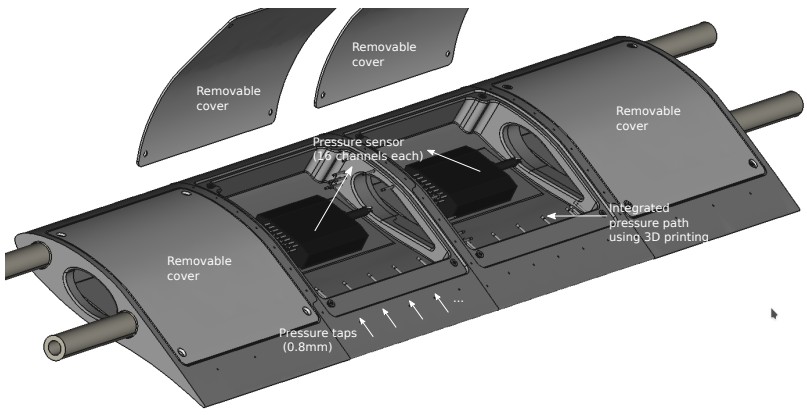

**Figure 8.** Computer Aided Design (CAD) drawing of the 3D printed 2D blade section mounted in the wind tunnel. The 2D blade section is made of aluminium. It is a hollow model that contains four pressure scanners with 16 channels each, which are connected to integrated pressure taps. The 2D blade section has four removable covers on the suction side to grant access to the pressure scanners.

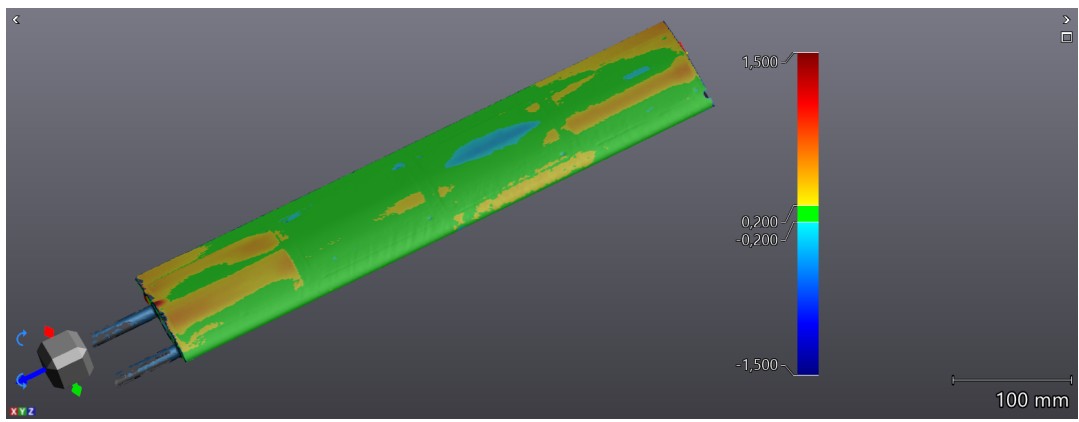

**Figure 9.** Illustration of the deviations (in mm) between the airfoil's original design shape and the shape achieved after the manufacturing process.

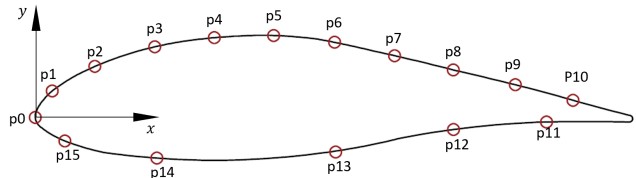

**Figure 10.** Chord-wise distribution of pressure ports around the airfoil - note that the axes are not scaled correctly.

blade designs, and Tygon tubes. Three chord-wise lines of pressure taps were distributed (between the covers): at the mid-span, $z/c = 0$, and at $z/c = \pm 1$. The chord-wise distribution is identical for the three lines, and it is shown in figure 10 with exact positions in table 1. One span-wise line of pressure taps was added at $x/c = 0.88$, with pressure taps spaced by $0.16c$ from $z/c = -1$ to $z/c = 1$ and $0.25c$ otherwise.

**Table 1.** Position of pressure ports

| Ports | p0 | p1 | p2 | p3 | p4 | p5 | p6 | p7 | p8 | p9 | p10 | p11 | p12 | p13 | p14 | p15 |
|---|---|---|---|---|---|---|---|---|---|---|---|---|---|---|---|---|
| $x/c$ | 0 | 0.03 | 0.1 | 0.2 | 0.3 | 0.4 | 0.5 | 0.6 | 0.7 | 0.8 | 0.9 | 0.85 | 0.7 | 0.5 | 0.2 | 0.05 |

### 4.1.3   Global load sensors

The airfoil was supported on two sides by load cells, cf. figure 11. The load cells work by the principle of strain measurement. Two load cells (one on each side) were used to measure the lift force ($F_l$), and two were used to measure the drag force ($F_d$). The load measurement system was calibrated in all directions (i.e., $+x$, $-x$, $+y$, and $-y$) using calibration weights between 500 g and 5000 g. The angle of attack was measured using a high-precision voltage-based angle sensor with a resolution of $0.035°/$mV. The signal from each load cell was collected at a sampling frequency of $f_s = 5000$ Hz. The lift coefficient ($C_l$) and the drag coefficient ($C_d$) were calculated using equations 40, and41,

$$C_l = \frac{F_l}{\frac{1}{2}\rho A U^2},$$

(40)

$$C_d = \frac{F_d}{\frac{1}{2}\rho A U^2},$$

(41)

where $\rho$ is the density of air, $A = 0.125$ m$\times$ 0.5 m is the model planform area, $U$ is the inflow speed and $\nu$ is kinematic viscosity.

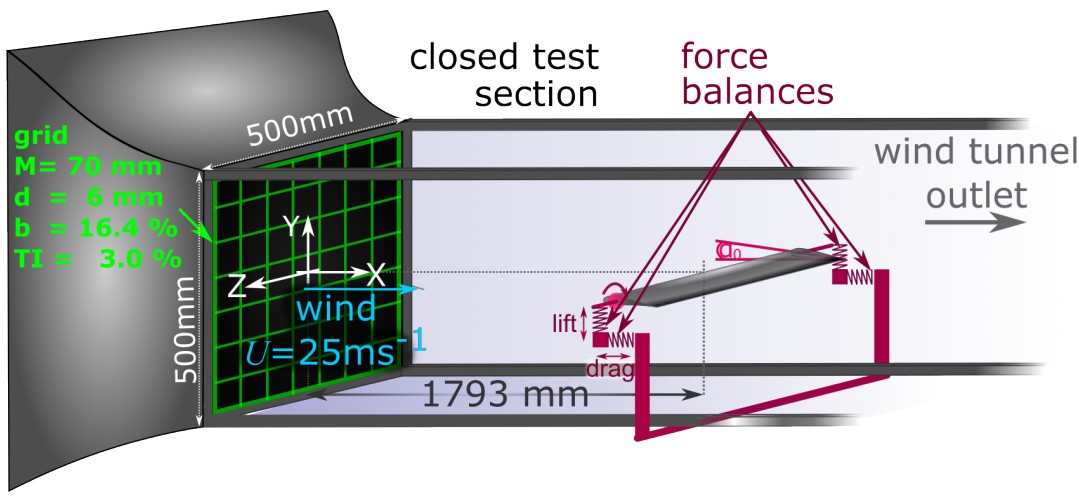

**Figure 11.** Wind tunnel set-up for load measurements.

## 4.2 Numerical set-up for airfoil simulations

Figures 12(a), (b), and (d) present the 3D view, front view, and side view of the numerical domain (including mesh), respectively, for an angle of attack of the 2D blade section of $\alpha=0°$. The mesh consists of 3 million cells, and figure 12(c) displays a close-up of the mesh around the airfoil. In this study, we only simulated the transverse half of the wind tunnel, as shown in figure 12(a). The shaded area in the figure was not simulated; instead, we applied a mirror boundary condition. The chord length of the airfoil is 0.125 m, which is the same as that used in the experiments. The airfoil is positioned at a downstream distance of 1.035 m from the domain's inlet, as the simulation domain starts at the position of the first measurement point, where the inlet conditions are obtained and from where the turbulence decays. Therefore, the downstream position of the airfoil in the simulation is 0.758 m less than that in the experiments, resulting in a downstream position of 1.035 m.

We have applied a Dirichlet boundary condition at the inlet, and the values are given in table 2. These values correspond to the values obtained at $x/M = 11$ from the hot-wire measurement (see section 3.4). For pressure, we applied the Neumann boundary condition at the inlet, $\frac{dp}{dn} = 0$, where $n$ is the normal vector to the inlet. These same values have been used as the initial conditions as well. In addition, we also use the integral length scale ($L = 25$ mm) to investigate the impact of using the "wrong" length scale at the simulation domain inlet. At the outlet, the velocity is found using Rhie and Chow interpolation. We applied the Dirichlet boundary condition at the outlet for pressure $p = p_o$, where $p_o = 0$ by default. For TKE and turbulence frequency, we applied the Neumann boundary condition as $\frac{dk}{dn} = 0$ and $\frac{d\omega}{dn} = 0$, respectively. We have applied a no-slip boundary condition on the airfoil, and imposed wall functions

$$\frac{\partial U}{\partial y} = \frac{\tau_s}{\kappa \rho c_\mu^{1/4} \sqrt{k_w} y_w}, \tag{42}$$

on the top wall (TW), bottom wall (BW), and side wall (SW) to avoid explicitly simulating the boundary layer. Here, $U$ is the velocity, $k_w$ is the TKE at the cell centre of the first cell from the wall, $y_w$ is the perpendicular distance of the cell centre of

the first cell from the wall, $\tau_s$ is the wall shear stress, $\kappa = 0.41$, and $c_\mu = 0.09$. The $y+$ values for airfoil, BW, TW, and SW is given in the table 3.

**Table 2.** Boundary conditions at the simulation domain inlet. Note that the for the inlet length scale $L_S$, the Taylor micro-scale is used.

| Variable | Value at inlet |
|----------|----------------|
| $U$ | $25 \text{ ms}^{-1}$ |
| $k$ | $1.859 \text{ m}^2\text{s}^{-2}$ |
| $\omega$ | $657.4 \text{ s}^{-1}$ |
| $L_S$ | $2.54 \text{ mm}$ |

**Table 3.** $y^+$ values for the simulation.

| Boundary | Applied $y^+$ | Average $y^+$ | $y^+$ Range |
|----------|---------------|----------------|-------------|
| Airfoil | 0.15 | 0.05 | 0.01 - 0.30 |
| Top Wall (TW) | 50 | 15 | 0.5 - 30 |
| Bottom Wall (BW) | 50 | 15 | 0.5 - 30 |
| Side Wall (SW) | 1 | 2 | 0.2 - 12 |

We conducted unsteady 3D RANS simulations using the $k-\omega$ SST Menter 2003 model (Menter et al., 2003) following the same procedure and using the same equations that we presented in the section 3.2. We performed these simulations using a standard AVLSMART numerical scheme. To improve the computational efficiency and to accurately capture the flow downstream of the airfoil, we utilized an in-house adaptive grid refinement (AGR) methodology (Wackers et al., 2012). The total number of cells at the end of the simulation was approximately 16 million (see figure 12(e)).

To obtain well-converged results for each angle of attack (AoA), we conducted 3D simulations using 400 cores on the IDRIS Jeay Supercomputer. This process took almost 100 computing hours for each AoA.

### 4.3 Comparison between experiments and simulations

In the following sections, we present a comparison between the force coefficients and pressure coefficients obtained from the experiments and simulations.

#### 4.3.1 Comparison of force coefficients obtained from digital twin and experiments

Figures 13 and 14 present a comparison between the $C_l$ and $C_d$ curves derived from wind tunnel experiments and their digital counterpart for a Reynolds number of $2.0 \times 10^5$. In one case, the inlet length scale for the simulation domain is defined by $\lambda$, while in the other case, it is determined by the integral length scale. The experimental tests covered angles of attack (AoAs) ranging from $-5°$ to $24°$, whereas the simulations were conducted for AoAs between $0°$ and $20°$. The figures clearly

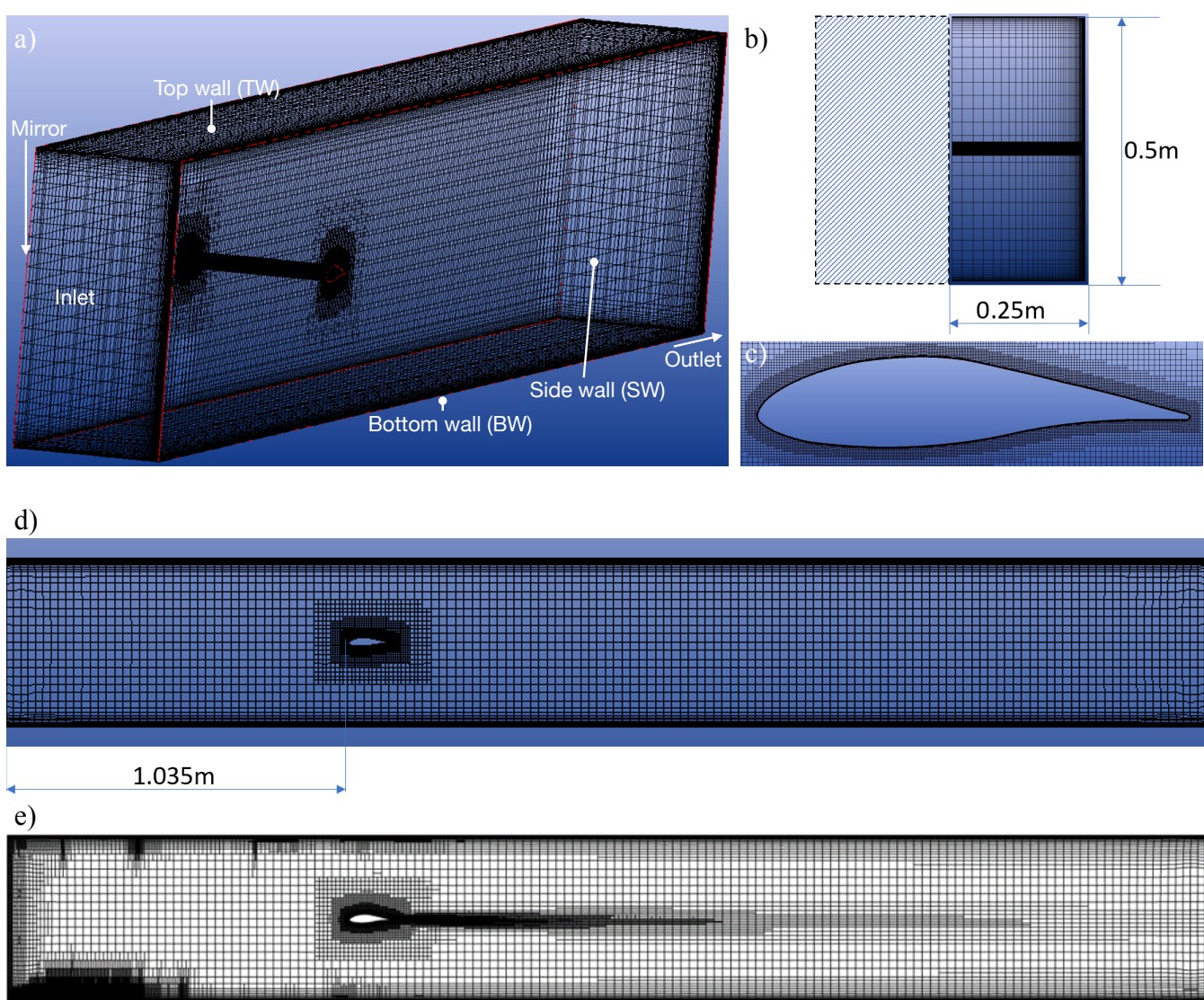

**Figure 12.** Views of the mesh: (a) 3D mesh for the simulation of the flow over the airfoil (3 million cells); (b) Side view of the mesh; (c) Mesh around the airfoil; (d) Front view of the mesh; (e) Side view of the mesh at the end of simulation after adaptive grid refinement (16 million cells).

demonstrate that our digital twin, utilizing $\lambda$ as the inlet length scale for the simulation domain, successfully replicates the force coefficients obtained in the original wind tunnel experiments under turbulent inflow conditions. In contrast, when the integral length scale is used as the inlet length scale, higher lift values are observed across all AoAs (with the exception of $0°$), with the disparity increasing as the AoA increases. This increase in lift coefficient values in the simulations, utilising $L$ as the boundary condition, can be attributed to the greater turbulence intensity experienced by the airfoil in the simulations compared to the experiments (cf., Abbott and Von Doenhoff (2012)). The inadequate representation of the evolution of the turbulent kinetic energy in the simulations subsequently affects the representation of the turbulence intensity, resulting in notable deviations in the lift coefficients ($C_l$) between the simulations and experimental data.

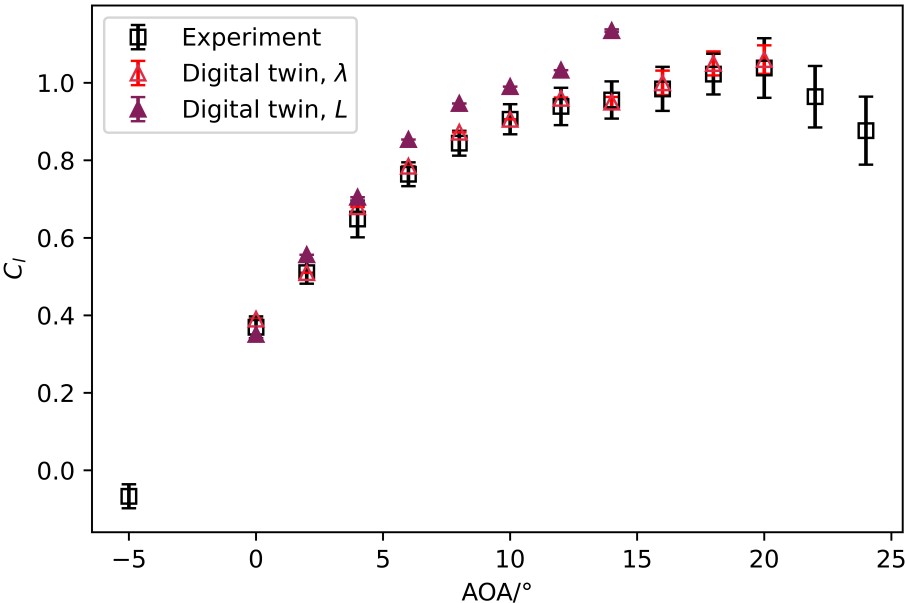

**Figure 13.** Comparison of $C_l$ curves obtained from digital twin (using Taylor micro-scale ($\lambda$) and integral length ($L$)) and experiments for $Re_c = 2.0 \times 10^5$. Error bars presented in the figure are the standard deviation of the time series data.

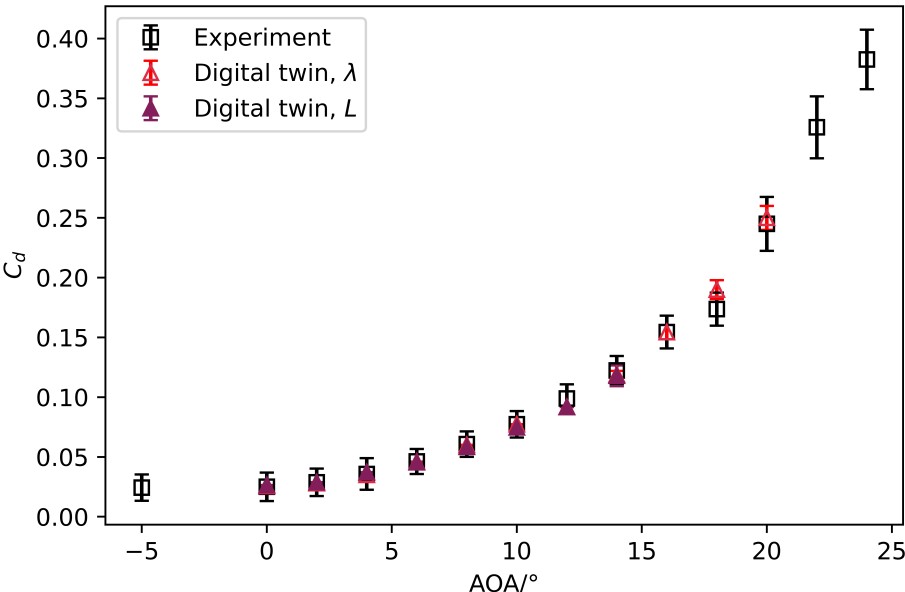

**Figure 14.** Comparison of $C_d$ curves obtained from digital twin (using Taylor micro-scale ($\lambda$) and integral length ($L$)) and experiments for $Re_c = 2.0 \times 10^5$. Error bars presented in the figure are the standard deviation of the time series data.

In the next subsection, a comparison of the $C_p$ obtained from simulations and experiments is made.

### 4.3.2   Comparison of $C_p$

Pressure measurements were performed over the surface of the airfoil, both experimentally and in the digital twin, for different angles of attack (AoAs). Specifically, we compared the $C_p$ values at AoAs of 0°, 4°, and 12°, encompassing a broad range of angles. For that, we average over three span-wise positions ($z = 0$ mm, $z = \pm125$ mm) and plot the average. Figures 15, 16,
and 17 depict the comparison between experimental data and 3D simulation results. In these comparisons, we utilized both the Taylor length scale and the integral length scale as inlet conditions for the simulation domain.
Overall, the simulation results closely align with the experimental results. However, a slight tendency toward higher pressure on the pressure side is evident. The exact cause of this variation remains unknown at present; differences in the extraction of the reference pressure and the dynamic pressure in simulations and experiments might contribute to these variations.
Moreover, the differences in $C_p$ levels between using the Taylor length scale and the integral length scale become more pronounced as the angle of incidence increases. These differences primarily manifest on the suction side of the airfoil, particularly in the region of the leading-edge suction peak.

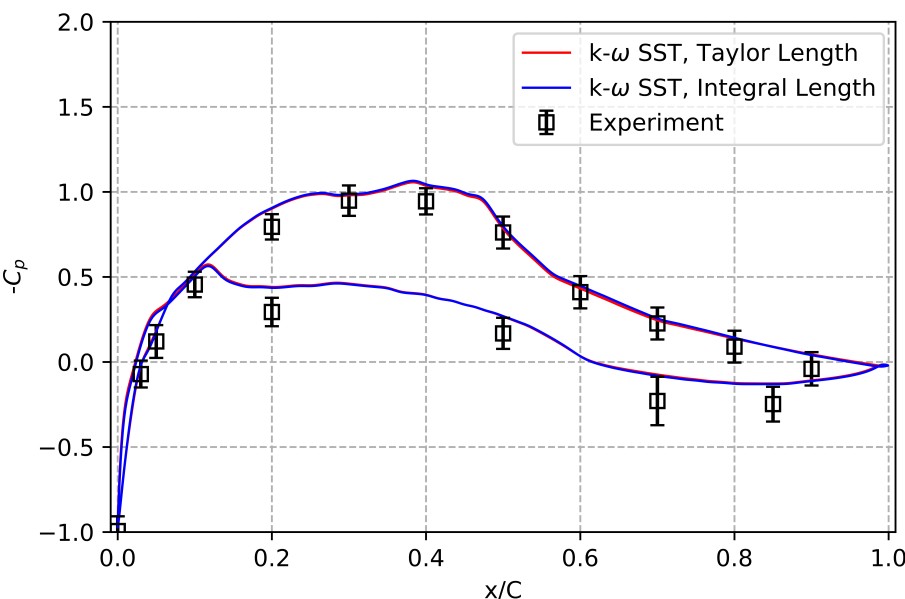

**Figure 15.** Comparative analysis of the $C_p$ curve for AoA = 0° and $Re_c = 2.0 \times 10^5$, considering 3D simulations and experimental data. The comparison is conducted by averaging the results obtained from three span-wise positions ($z = 0$ mm, $z = \pm125$ mm).

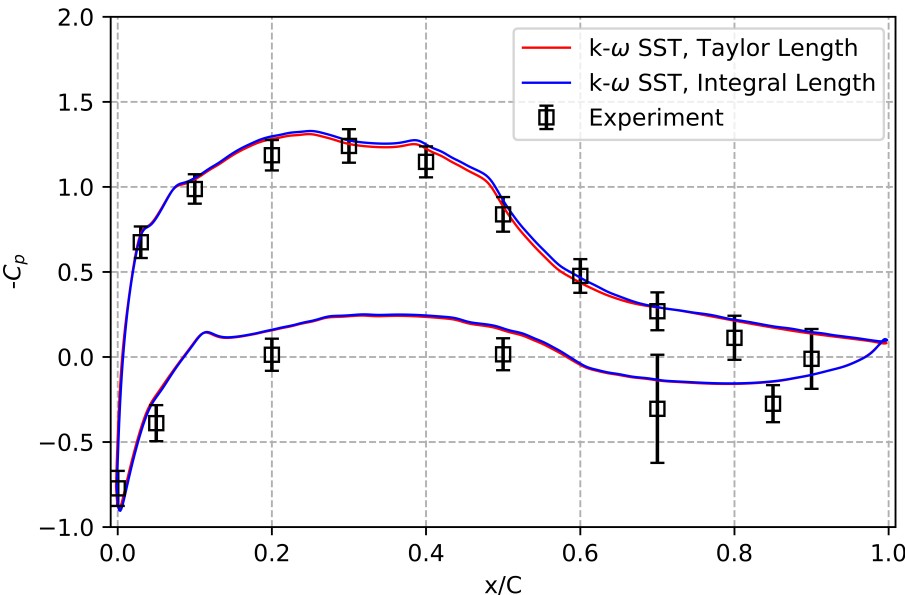

**Figure 16.** Comparative analysis of the $C_p$ curve for AoA = 4° and $Re_c = 2.0 \times 10^5$, considering 3D simulations and experimental data. The comparison is conducted by averaging the results obtained from three span-wise positions ($z = 0$ mm, $z = \pm125$ mm).

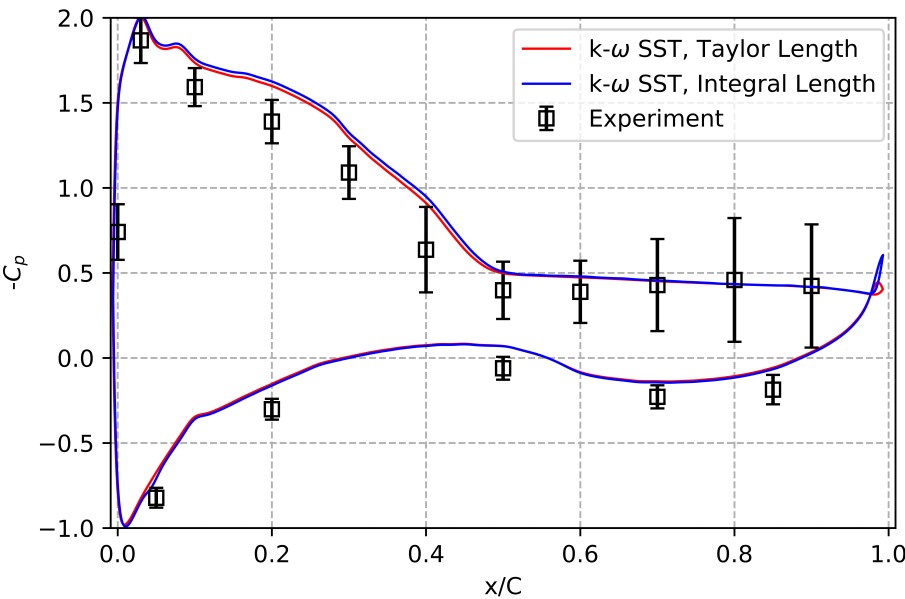

**Figure 17.** Comparative analysis of the $C_p$ curve for AoA = 12° and $Re_c = 2.0 \times 10^5$, considering 3D simulations and experimental data. The comparison is conducted by averaging the results obtained from three span-wise positions ($z = 0$ mm, $z = \pm 125$ mm).

### 4.3.3 Comparison of the performance of 2D digital digital twin against 3D digital twin

This section provides a comparative study of force coefficients derived from both 2D and 3D digital twin simulations. Consistent initial and boundary conditions were maintained in both models. Figures 18 and 19 visually display the comparison for the lift and drag coefficient, respectively. The 2D outcomes are graphed for angles of attack between -5° and 16°, while the 3D results cover 0° to 16°.

The findings show a considerable correlation in the $C_l$ values until an AoA of 12°. The 2D digital twin exhibits higher $C_l$ values compared to the 3D digital twin for angles of attacks (AoAs) of 14° and above. The $C_d$ values show a close match across all AoAs, with the exception of 16° AoA, at which point the 2D digital twin displays a higher $C_d$ than its 3D counterpart.

The difference in $C_l$ values between the 3D and 2D digital twins can be attributed to the growing importance of 3D effects, for example flow bi-stability, which impacts the position of flow separation at and above 14° AoAs in high Reynolds number experiments at the same airfoil in Neunaber et al. (2022), something which cannot be reproduced in the 2D digital twin. As a result, if the emphasis is purely on force coefficients, the 2D digital twin can be used for low to moderately high AoAs due to its computational efficiency. For higher AoAs, where 3D effects are substantial, the more computationally demanding 3D digital twin is preferable. This strategic blend optimizes the use of the digital twin framework.

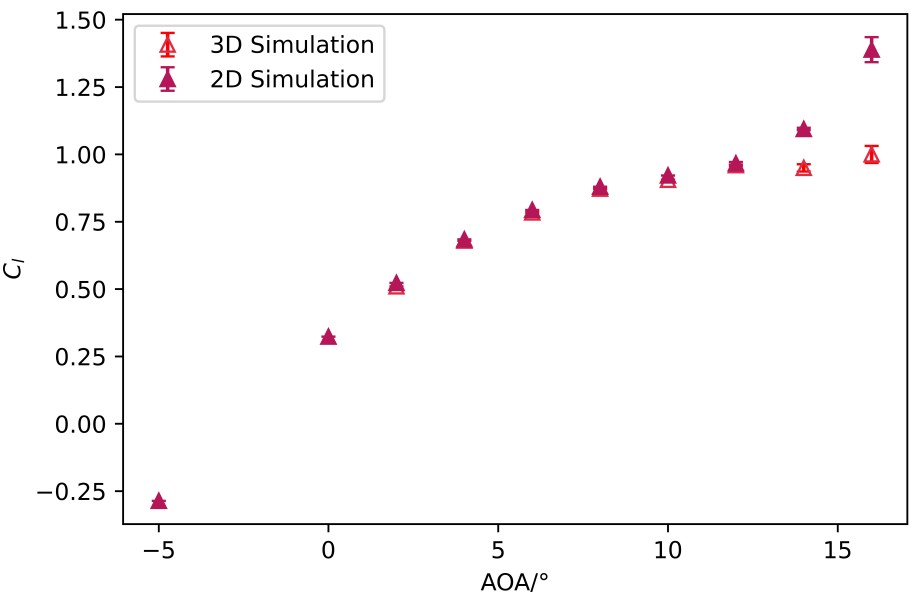

**Figure 18.** Comparison of the $C_l$ curve obtained from 3D digital twin and 2D digital twin for $Re_c = 2.0 \times 10^5$

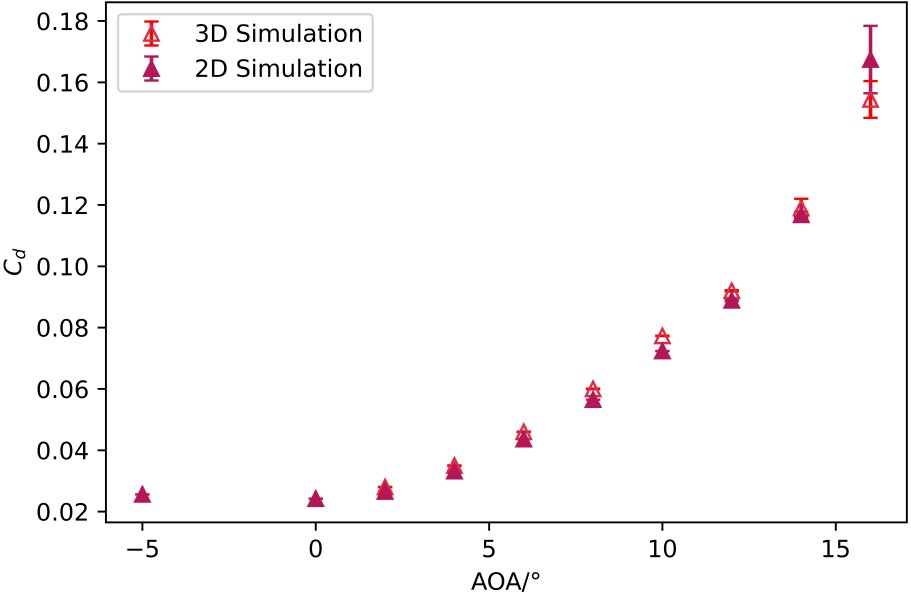

**Figure 19.** Comparison of the $C_d$ curve obtained from 3D digital twin and 2D digital twin for $Re_c = 2.0 \times 10^5$

# 5 Conclusions

We have successfully created a digital twin of a wind tunnel with turbulent inflow conditions after creating a theoretical framework for the decay of the TKE, and we successfully expanded this digital-twin to perform 3D simulations of the impact of turbulent inflow on a 2D blade section. In the first part, a theoretical proof of the dependence of the downstream evolution of the turbulent kinetic energy on the Taylor micro-scale was developed for regular-grid-generated turbulent flows. It has then been demonstrated that the Taylor micro-scale is the correct turbulent length scale to be used as the boundary condition at the inlet in RANS simulations for the accurate prediction of the downstream evolution of the turbulent kinetic energy. To validate our results, we compared the theoretical development with measurements performed downstream of a regular grid and RANS simulations using the $k - \omega$ SST Menter 2003 model. When the Taylor micro-scale (measured experimentally) is used as the inlet condition both for RANS simulations and for the starting point of the theoretical equations, the turbulent kinetic energy measured experimentally is retrieved. In contrast, when the integral length scale is used as the inlet boundary condition, RANS simulations are far from experiments and theory. Further, we compare the RANS results using several $k - \omega$ models with the Taylor micro-scale as the boundary condition, and the results are in good agreement with each other. This work thus demonstrates the validity of closure models in RANS equations to describe homogeneous, isotropic flows, as long as the Taylor micro-scale is used as one inlet boundary condition instead of the integral length scale. Further, our results emphasise fundamental behaviours of grid-generated turbulent flows:

1. The spatial evolution of the turbulent kinetic energy has a fundamental dependence on Taylor micro-scale, see equation (23).

2. For properly capturing the evolution of the turbulent kinetic energy either in space or time, the correct length scale given as the boundary condition is the Taylor micro-scale.

In the third part, we introduced an airfoil into the numerical wind tunnel and conducted 3D simulations at a chord-based Reynolds number of $Re_c = 2.0 \times 10^5$. Our findings showed a nearly perfect agreement between the force coefficients obtained from experiments in the physical wind tunnel and those obtained from simulations in the numerical wind tunnel when using the Taylor micro-scale ($\lambda$) as the simulation domain inlet length scale. This validates the suitability of the numerical wind tunnel for our purposes. In contrast, simulations using the integral length scale at the inlet boundary resulted in significant differences in lift coefficients when compared to the experimental results. This demonstrates that accurately capturing the evolution of the turbulent kinetic energy upstream of the airfoil is crucial for reproducing its aerodynamic behavior in numerical simulations. The comparison between the chord-wise pressure coefficients obtained from experiments and the digital twin revealed that they were similar on the suction side. However, significant differences in $C_p$ values were observed on the pressure side. As the force coefficients obtained using load cells matched well with those obtained in the digital twin, this suggests that there may be room for improvement in the pressure measurement experiments. Moving forward, we plan to perform simulations at higher Reynolds numbers of approximately $O(10^6)$ to further validate our numerical wind tunnel.

A comparison between 2D and 3D simulations also indicates that, if the emphasis is purely on force coefficients, the 2D digital twin can be used for low to moderately high AoAs due to its computational efficiency. For higher AoAs, where 3D effects are substantial, the more computationally demanding 3D digital twin is preferable. This strategic blend optimizes the use of the digital twin framework.

The authors advise the simulation community to be aware of the significant impact that the Taylor micro-scale ($\lambda$) has on the dissipation rate, which in turn affects the evolution of turbulent kinetic energy (TKE). Although different simulation codes may define length scales differently, they should all produce the same dissipation rate. To ensure consistency, the boundary condition for length scales in simulations should either be the Taylor micro-scale itself or a function of it (depending on the definition used in a particular code). In the authors' study, the Taylor micro-scale is used directly as the boundary condition.

The methodology presented in this paper is tailored to regular-grid-generated turbulent flows within a specific scope. As a next step, we intend to investigate anisotropic flows, particularly shear flows, in order to gain insights into the downstream and transverse evolution of turbulent kinetic energy. Our goal is to extend equation (23), currently formulated in 1D, into its 2D counterpart. In future work, we plan to conduct simulations at higher Reynolds numbers, around $Re_c \sim O(10^6)$, to validate our numerical wind tunnel in the context of elevated Reynolds numbers.

*Data availability.* The datasets examined in this study are available at the following DOI link: https://dx.doi.org/10.25326/554

## Appendix A: Details of experiments performed at University of Oldenburg

The experiments from the University of Oldenburg were carried out in the large wind tunnel that has an inlet of $3 \times 3$ m and a test section length of 30 m. A single hot-wire was operated using a StreamLine 9091N0102 frame with a 91C10 CTA (Constant Temperature Anemometry) Module. It was sampled at $f_s = 20$ kHz; a hardware low-pass filter with a cut-off frequency of $f = 10$ kHz was set. The experiments were performed using a passive regular grid with $M = 115$ mm, and 33 downstream positions spanning from $x/M \sim 8$ to $x/M \sim 170$ were traversed for two inflow speeds: 5 ms$^{-1}$ and 10 ms$^{-1}$. The experimental data was obtained at the span-wise centre at the height of $8M$ above the floor of the wind tunnel.

*Author contributions.* RM developed the theoretical framework, carried out the simulations, acquired the aerodynamic data, performed the initial analysis and data investigation and wrote the original draft. IN acquired the hot-wire data at Centrale Nantes and in Oldenburg and performed the initial analysis and data investigation. CB prepared the blade design with wall pressure measurements, followed its manufacturing and its shape corrections to match simulation results. CB acquired the funding. CB, EG, RM and IN developed the methodology. CB, EG and IN reviewed and edited the manuscript. EG supervised all the simulations.

*Competing interests.* The contact author has declared that neither they nor their co-authors have any competing interests.

*Acknowledgements.* This work is funded under French national project MOMENTA (grant no. ANR-19-CE05-0034) and the computations were performed using HPC resources from GENCI (Grand Equipement National de Calcul Intensif) (Grant-A0132A00129) which is gratefully acknowledged. The blade was manufactured thanks to the ePARADISE project with the funding from ADEME and Pays-de-Loire region in France (grant no. 1905C0030). Part of the measurements have been performed during a stay associated with a twin-fellowship from the Hanse-Wissenschaftskolleg (HWK Institute for Advanced Study, Delmenhorst, Germany) assigned to Ingrid Neunaber. We would like to thank Prof. Dr. Martin Obligado and Dr. Michael Hölling for their collaboration for these measurements.

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
