# Peer review of "Developing a Digital Twin Framework for Wind Tunnel Testing: Validation of Turbulent Inflow and Airfoil Load Applications"

_Wind Energy Science, 2023_

## Author Response (AR1)

**Comments from the reviewer 1**

The manuscript discusses the use of a digital twin to perform wind turbine and airfoil tests in wind tunnels with turbulent inflows. Hot-wire anemometry is first used to characterize grid-generated turbulence in the wind tunnel. Pressure and force sensors are then applied to record the pressure, drag and lift coefficients of an airfoil under different angles of attack at a chord-based Reynolds number of $2.0x10^5$. On the other hand, RANS simulations at the same Reynolds number are used to capture the kinetic energy decay of the grid-generated flows and the different coefficients within the airfoil. Good agreement is found between all quantities, provided that the Taylor microscale is used in the RANS simulations as the length scale required to simulate the grid-generated turbulent flow.

I find the thematic of this manuscript within the scope of the journal. Furthermore, it is well written and organized, with a theoretical discussion and numerical results that are of interest for the wind energy community. Nevertheless, before recommending publication the authors should assess the following points:

**Response**: We are grateful for the reviewer's remarks on our manuscript that helped to improve its quality. The issues they pointed out have been duly addressed, and corresponding adjustments have been made in the updated version of the manuscript.

**Comment 1:** Several arguments used to deduce equations 23 and 27 rely on the presence of a fully developed grid-generated turbulent flow, that is only found far downstream the grid. The range studied here ($x < 30M$, with the airfoil placed at $x \sim 20M$) may present some differences in terms of the approximations made in equations 10, 13 and 20. While the authors refer to a previous publication from the group, these points should be addressed in the present manuscript.

> **Response 1:** The authors would like to thank the reviewer for mentioning this important point. In response we have added the following statement in the section 2.3 line 245
>
> "The only assumption taken while deriving the equation (23) is statistical stationarity of fully developed GGT which is believed to start from $x/M \approx 20$ downstream of a grid (Comte-Bellot and Corrsin, 1966; Bailly and Comte-Bellot, 2015). Upstream of $x/M \approx 20$, one may expect some changes in the form of equations (10), (13), and (20) which are used to derive equation (23). However, between $x/M \approx 10$ and $x/M \approx 20$ the turbulent flow can be considered approximately developed (Frisch, 1995).Therefore, for all practical purposes, equation (23) is valid downstream of $x/M \approx 10$."
>
> Following the reviewer's remark, we extended the application of equation (23) to additional data sets with experimental measurements exceeding $x > 30M$. This comparison are illustrated in figure 1. It can be clearly seen that equation (23) matches very well with the experimental data. Note that the deviations are over-accentuated when visualised in the log-log plot. More details on this comparison are also presented in the section 3.4. of the manuscript.

**Comment 2:** Related to my previous point, several papers discuss the decay of kinetic energy in terms of invariants (Sinhuber et al, PRL 2015; Krogstad and Davidson JFM 2009), and also the role of the integral length scale on such models. Furthermore, the power laws predicted contain a virtual origin that is not present in the theoretical discussion of the present work. First, these approaches should be mentioned at the section 'Brief theoretical description of decaying grid turbulence'. Second, they should be at least addressed when figure 4 is presented. Have

[Figure]

Figure 1: Comparison of experimental decay of normalised TKE decay obtained from experiments performed in the University of Olden- burg, and data from Batchelor and Townsend (1948) (B&T) with equation (23). For the application of equation (23) to the B&T data, we estimate $\lambda \approx 9$ mm from their manuscript.

the authors tried to compare their data with them? Are the decay exponents near the ones predicted in such papers?

**Response 2:** The authors would like to thank the reviewer for suggesting these interesting papers. In response to this question, we have added the following sentence in the section 'Brief theoretical description of the decaying turbulence' line 60:

"Krogstad and Davidson (2009) showed that grid turbulence is Saffman turbulence (Saffman, 1967). They improved the decay exponent of TKE from 1.2, which Saffman gave for perfectly homogeneous isotropic turbulence, to 1.1 for GGT. Sinhuber et al. (2015) performed experiments with one grid for different Reynolds numbers ($Re_M = \frac{UM}{\nu}$, where $M$ is the grid mesh size, and $U$ the mean velocity) and found that the decay exponent of TKE was equal to 1.18. They also showed that the decay exponent was independent of $Re_M$."

On the reviewer's suggestion, we did compare our decay exponent with the decay exponents presented in the suggested papers and have added the following sentence in the section 3.4 line 345 where figure 4 is presented:

" The TKE's decay exponent was determined to be 1.087 in the equation (23) and simulation, while in the experimental data, it was observed to be 1.09. These values are in close proximity to the Saffman decay exponent of 1.1 for grid-generated turbulence, as reported by Krogstad and Davidson (2009)"

We have also made it explicitly clear that the equation (23) does not contain any fitting parameters including a virtual origin. We have added the following statement in section 2.3 line 242:

"The authors wish to emphasise to the readers that, unlike the equations commonly encountered in prior literature, such as those referenced in Comte-Bellot and Corrsin (1966), Kurian and Fransson(2009), Krogstad and Davidson (2009), or Sinhuber et al. (2015), equation (23) does not have any fitting parameter, and it is neither an empirical equation nor does it have any virtual origin."

**Comment 3:** Presenting figure 4 in logarithmic scale (or using an inset for that) would help to assess the quality of the power-law adjustment. Also, if I understand correctly, the theoretical curve has no fitting parameters? If that is the case, it should be made more explicit on the text as it is a relevant result.

**Response 3:** Figure 4 has been revised to include the plot also with a logarithmic scale. Indeed, the theoretical curve has no fitting parameters, which is now emphasized in the manuscript 2.3.

**Comment 4:** To properly address the relevance of the Taylor scale on the RANS models, a more systematic study with several grids (producing different values of integral and Taylor scales) should be performed. The present study is an interesting contribution pointing towards the relevance of the Taylor scale in RANS modelling, but gives no conclusive proof. I consider that the conclusions should emphasize this point.

**Response 4:** We see the reviewer's concern, and we would like to discuss how figure 1 shows the generality of equation (23) as it matches very well with different experimental data obtained for different grid sizes and inflow velocities. Equation (23) and the TKE decay equation derived within the $k - \omega$ framework, equation (27), are equivalent and give the same decay. It can therefore be understood that $k - \omega$ models will also give the same decay as equation (23) provided we use Taylor micro-scale as inlet length scale. Thus, this shows the generality of using the Taylor micro-scale as inlet length scale in RANS simulations to get the proper downstream TKE decay for GGT.

**Comment 5:** The accuracy of the numerical simulation in the estimation of different coefficients is discussed in terms of the experimental results but not compared with other numerical works/schemes. I suggest that the authors discuss other works from the literature when presenting figures 16 to 22. Also, those figures should have error bars added.

**Response 5:** As far as the authors are aware, there has been no previous numerical study conducted on this particular airfoil. It makes us unable to compare the obtained lift and drag coefficients with other numerical studies. The simulations presented in this paper were executed using the standard AVLSMART numerical scheme. In response to the reviewer's recommendation, we conducted an additional simulation at the 14° angle of attack (AoA) employing the BLENDED numerical scheme. However, we observed no disparity in the values of lift and drag coefficients. Error bars are now added to the figures

**Comment 6:** I also suggest that figures 11 to 15 are merged onto a single one.

**Response 6:** Following the reviewer's suggestion we have combined figures 11 to 15 into a single figure.

**Comments from the reviewer 2**

**1 General comments**

**Comment:** This manuscript is an interesting study on grid generated turbulence, its modelling, and its effect on aerodynamic forces in the flow around an airfoil. The topic is investigated using a combination of theoretical, computational, and experimental approaches, and the results and conclusions are of interest of the readership of Wind Energy Science and the fluid mechanics community in general. The recommendation of using the Taylor micro-scale to define RANS inlet boundary conditions is very useful for high-fidelity modelling in wind energy.
So, the work constitutes a contribution that deserves to be published, but there are many corrections and improvements that need to be made before. They are listed below.

**Response:** We thank the reviewer for their encouraging comments on our manuscript that helped improve its quality. The highlighted issues have been addressed, and the necessary changes have been incorporated into the revised manuscript.

**2 Specific comments**

**Comment 1:** Page 2, line 25: The manuscript says "Alternatively, experiments can be conducted in a wind tunnel by subjecting a Reynolds-scaled wind turbine rotor or a blade section from a real wind turbine blade to turbulent inflow under different inflow conditions, such as homogeneous inflow or gust inflow." Wind turbine rotor models for wind tunnel tests are usually not scaled according to the Reynolds number, because in this case, the wind speed in the tunnel would have to be prohibitively high. They usually obey Strouhal similarity (i.e., keep the same tip speed ratio). Some sort of measure has to be taken or assumption has to be made to account for this Reynolds number discrepancy. Roughness elements can be added to the blades, or it the experiments can be run in a way that it guarantees that the flow will be fully turbulent at the rotor blades boundary layer (wind speed or turbulent kinetic energy high enough). For rotor blade sections tested in fixed conditions, as the case of the tests reported in this manuscript, the situation is different. So, I believe this sentence should be rewritten.

**Response 1:** Indeed, for wind turbines, the Strouhal similarity is used for scaling down. Based on the suggestion, we have revised the statement to: "Alternatively, experiments can be conducted in a wind tunnel by subjecting a Reynolds-scaled blade section from a real wind turbine blade to turbulent inflow under different inflow conditions, such as homogeneous inflow or gust inflow".

**Comment 2:** Page 2, line 52: The manuscript says "The Navier-Stokes equations are highly nonlinear...". The Navier-Stokes equations have just one nonlinear term, and it is a quadratic term. In my point of view, this does not make them "highly" nonlinear. I suggest the authors remove the term "highly".

**Response 2:** We thank you for bringing this to our attention. We have removed the word "highly" and retained the term "nonlinear.

**Comment 3:** Page 2, line 53: Why do the authors use GDT as acronym for grid-generated turbulence? Would not be more appropriate to use GGT?

**Response 3:** Certainly, opting for GGT instead of GDT is not only more suitable but also enhances clarity. Consequently, we have replaced GDT with GGT.

**Comment 4:** Page 3, line 58: The authors should mathematically define TKE and dissipation at this point.

**Response 4:** We have now provided the mathematical definition of TKE and dissipation on Page 3, line 58.

**Comment 5:** Page 5, section 2.1: I believe the manuscript would benefit from an explanation of the physical meaning/definition of the Taylor micro-scale at this point. The authors should also mention how it could be calculated or measured.

**Response 5:** The authors extend their appreciation to the reviewer for their input. Physically, Taylor micro-scale represents small eddies (Batchelor and Townsend, 1948). This physical meaning of Taylor micro-scale is written in the line 144-145 of section 2.1: "Batchelor and Townsend (1948) proposed to call the Reynolds number based on the Taylor micro-scale ($\lambda$) the "Reynolds number of turbulence", and also suggested that $\lambda$ is representative of the eddies of large wave-number, i.e., small eddies, before viscosity becomes relevant.".

Given that the mathematical definition and experimental methodology for calculating $\lambda$ are presented in Section 3.3, the authors have incorporated the following sentence: "The mathematical definition and the experimental methodology for calculating $\lambda$ are detailed in section 3.3." in line 147 at page 5, section 2.1.

To improve the clarity the statement in line 328, page 13, section 3.3, the original statement "where $\langle(\frac{\partial u}{\partial x})^2\rangle$ can be estimated from the spectrum in the wave-number ($\kappa$) domain" is now improved to "where $\langle(\frac{\partial u}{\partial x})^2\rangle$ can be determined from the spectrum in the wave-number ($\kappa$) domain, derived through hot-wire measurements using Taylor's hypothesis"

**Comment 6:** Pages 10-11, section 3.2: The authors should present the equations that are solved in the RANS simulations, and the values of the constants that were employed in the turbulence quantities transport equations.

**Response 6:** The section 3.2 now contains the presentation of the transport equation for URANS, as well as the transport equations for turbulent kinetic energy ($k$) and specific dissipation or turbulence frequency ($\omega$), including the associated constant values.

**Comment 7:** Pages 12-13, section 3.4: The authors must run the same validation test for other wind speeds and add the results to this section.

**Response 7:** For regular grids with a given grid mesh size $M$ it is an established observation that the decay of normalized turbulent kinetic energy, expressed as $k/U^2$ (in our specific case, $k/0.5U^2$), when plotted against downstream distance normalized by the mesh size, denoted as $x/M$, remains independent of the inflow velocity $U$. This is very well documented in Batchelor and Townsend (1948) [1]. A relatively recent work from Sinhuber et al. PRL (2015) [3] also confirms this observation. Here the authors have performed experiments for Reynolds numbers ($Re_m$) ranging from $10^4$ to $5 \times 10^6$ for a given grid mesh size. On the, reviewer's recommendation, we performed simulations for $30\text{ms}^{-1}$ and $40\text{ms}^{-1}$ and we find the same evolution of the normalised decay of TKE for different inflow speeds (see

[Figure]

Figure 1: Normalised TKE decay for multiple velocities

figure 1).

Also, to perform an independent validation of equation (23), we tested it against hot-wire experiments performed in the wind tunnel at the University of Oldenburg, and data given in Batchelor and Townsend (1948). The experiments from Oldenburg were performed for a passive regular grid with $M = 115$ mm for 33 downstream positions spanning from $x/M \sim 8$ to $x/M \sim 170$. Two different inflow velocities, 5 ms$^{-1}$ and 10 ms$^{-1}$, were used. Figures 2 shows the log-log plot of the comparison of the TKE decay obtained experimentally with that from equation (23). It can clearly be seen that the evolution of TKE given by equation (23) matches very well with the experimental data. Note that that the deviations appear more pronounced when depicted in the log-log plot. We have included this supplementary validation in section 3.4.

**Comment 8:** Pages 17-19, section 4.2: The mathematical expressions of the boundary conditions must be presented (for velocity, pressure, and turbulence quantities). What was the initial condition used for the flow simulations? What were the wall functions employed at the top, bottom, and side walls? What were the value of y+ on those walls?

**Response 8:** We thank the reviewer for pointing this out. Following the reviewer's remark we have made following changes in the section 4.2:

We have added the following statement in response to the question on the inlet boundary condition and the inlet condition, also rephrasing the description of the values of $y+$ to make it clearer:

"We have applied a Dirichlet boundary condition at the inlet, and the values are given in table 1. These values correspond to the values obtained at $x/M = 11$ from the hot-wire measurement (see section 3.4). For pressure, we applied the Neumann boundary condition at the inlet, $\frac{dp}{dn} = 0$, where $n$ is the normal vector to the inlet. These same values have been used as the initial conditions as well. In addition, we also use the integral length scale ($L = 25$ mm) to investigate the impact of using the "wrong" length scale at the simulation domain inlet. At the outlet, the velocity is found using Rhie and Chow interpolation. We applied the Dirichlet boundary condition at the outlet for pressure $p = p_o$, where $p_o = 0$ by default. For TKE and turbulence frequency, we applied the Neumann boundary condition as $\frac{dk}{dn} = 0$ and $\frac{d\omega}{dn} = 0$, respectively. We have applied a no-slip boundary condition on the

[Figure]

Figure 2: Comparison of experimental decay of normalised TKE decay obtained from experiments performed in the University of Oldenburg, and data from [1] (B&T) with equation (23). For the application of equation (23) to the B&T data, we estimate $\lambda \approx 9$ mm from their manuscript.

airfoil, and imposed wall functions

$$\frac{\partial U}{\partial y} = \frac{\tau_s}{\kappa \rho c_\mu^{1/4} \sqrt{k_w} y_w},$$ (1)

on the top wall (TW), bottom wall (BW), and side wall (SW) to avoid explicitly simulating the boundary layer. Here, $U$ is the velocity, $k_w$ is the TKE at the cell centre of the first cell from the wall, $y_w$ is the perpendicular distance of the cell centre of the first cell from the wall, $\tau_s$ is the wall shear stress, $\kappa = 0.41$, and $c_\mu = 0.09$. The $y+$ values for airfoil, BW, TW, and SW is given in the table 2.

Table 1: Boundary conditions at the simulation domain inlet. Note that the for the inlet length scale $L_S$, the Taylor micro-scale is used.

| Variable | Value at inlet |
|----------|----------------|
| $U$ | 25 ms$^{-1}$ |
| $k$ | 1.859 m$^2$s$^{-2}$ |
| $\omega$ | 657.4 s$^{-1}$ |
| $L_S$ | 2.54 mm |

Table 2: $y^+$ values for the simulation.

| Boundary | Applied $y^+$ | Average $y^+$ | $y^+$ Range |
|----------|---------------|---------------|-------------|
| Airfoil | 0.15 | 0.05 | 0.01 - 0.30 |
| Top Wall (TW) | 50 | 15 | 0.5 - 30 |
| Bottom Wall (BW) | 50 | 15 | 0.5 - 30 |
| Side Wall (SW) | 1 | 2 | 0.2 - 12 |

**Comment 9:** Page 21, line 400: The manuscript says "However, there is a slight tendency towards higher pressure on the pressure side, which could be attributed to difficulties in defining the reference pressure (which was measured inside the blade).". What data shows this tendency? I guess it is the experimental set, but this should be stated clearly in the text. If the reason is difficulties in defining the reference pressure, why does not that affect the other measurement points?

**Response 9:** The authors recognise that the statement is a bit misleading as we do not fully understand what is causing this difference between $C_p$ curves obtained from our digital twin and experiments. However, we do expect that differences in the extraction of reference pressure and the dynamic pressure in simulations and experiments may lead to these differences. Further investigations are required to address factors that extend beyond the scope of this paper and cannot be solely explained by force coefficient measurements. We have replaced that statement with the following statement: "However, a slight tendency toward higher pressure on the pressure side is evident. The exact cause of this variation remains unknown; nonetheless, differences in extraction of reference pressure and dynamic pressure in simulations and experiments might contribute to these variations."

**Comment 10:** Page 21, line 404: The sentence "Nevertheless, these differences are not significant for these local quantities." is unclear. What are the authors trying to say?

**Response 10:** Following the changes in response to Comment 9 authors have removed this line from the paper.

**Comment 11:** Page 23, Figure 20: I believe the caption of the figure is wrong, because that graph probably refer to an angle of attack other than 0°, since results for that AoA are already shown in Figure 18.

**Response 11:** Thank you for pointing out this typo. The figure is associated to the AoA 12° and the caption of the figure is corrected

**Comment 12:** Page 23, line 414: "The discrepancy in Cl values between the 3d and 2d digital twins can be traced back to the escalating significance of 3d effects at and above 14° AoAs, which are unaccounted for in the 2d model." What 3d effects are those? It is necessary to provide a clear physical explanation here.

**Response 12:** Experiments performed on this airfoil at high Reynolds number have shown that span-wise flow organisation is highly correlated in space and time at AoAs 14° and more, leading to bi-stability and affecting the position of flow separation(see, Neunaber et al. (2022) [2]). Since, by design, 2D simulations cannot simulate these 3D effects, we see differences in $C_l$ between 2D and 3D.

We have updated the statement mention in the comment to the following: "The difference in $C_l$ values between the 3D and 2D digital twins can be attributed to the growing importance of 3D effects, for example flow bi-stability, which impacts the position of flow separation at and above 14° AoAs in high Reynolds number experiments at the same airfoil in neunaber2022wind, something which cannot be reproduced in the 2D digital twin."

**3    Technical comments**

- Page 2, line 30: Change "Veers et al. (2019)" to "(Veers et al., 2019)". The same correction applies to line 90.

- Page 2, line 31: Change "millions" to "million".

- Page 2, line 34: Change "a key" to "key" (in this sentence, key is an adjective, not a noun).

- Page 3, line 57: Change "(Kurian and Fransson (2009))" to "(Kurian and Fransson, 2009)". This type of mistake happens in many other places of the manuscript (lines 27-28, 73, 74, 75, 91 . . . ), please perform a thorough check.

- Page 3, line 62: Change "computational fluid mechanics" to "computational fluid dynamics".

- Page 3, line 79: Delete "have".

- Page 4, line 97: Change "a perfect" to "an adequate".

- Page 5, line 136: Delete "on".

- Page 5, line 145: Change "This is confirmed" to "This was confirmed".

- Page 8, line 210: Remove paragraph indentation.

- Page 11, line 273: Change "the ones" to "those".

- Page 13, figure 4: Add "(23)" after "Equation" in the figure legend.

- Page 17, line 353: Change "airfoil's cross section" to "model planform area".

**Response:** The authors express their gratitude to the reviewer for providing technical corrections. All suggested amendments have been fully integrated into the updated manuscript.

**References**

[1] George Keith Batchelor and Albert Alan Townsend. Decay of isotropic turbulence in the initial period. *Proceedings of the Royal Society of London. Series A. Mathematical and Physical Sciences*, 193(1035):539–558, 1948.

[2] Ingrid Neunaber, Frédéric Danbon, Antoine Soulier, Dimitri Voisin, Emmanuel Guilmineau, Philippe Delpech, Sébastien Courtine, Claire Taymans, and Caroline Braud. Wind tunnel study on natural instability of the normal force on a full-scale wind turbine blade section at reynolds number 4.7· 106. *Wind Energy*, 25(8):1332–1342, 2022.

[3] Michael Sinhuber, Eberhard Bodenschatz, and Gregory P Bewley. Decay of turbulence at high reynolds numbers. *Physical review letters*, 114(3):034501, 2015.